# TMAO, a seafood-derived molecule, produces diuresis and reduces mortality in heart failure rats

Marta Gawrys-Kopczynska[1], Marek Konop[1], Klaudia Maksymiuk[1], Katarzyna Kraszewska[1], Ladislav Derzsi[2], Krzysztof Sozanski[2], Robert Holyst[2], Marta Pilz[2], Emilia Samborowska[3], Leszek Dobrowolski[4], Kinga Jaworska[1], Izabella Mogilnicka[1], Marcin Ufnal[1]*

[1]Department of Experimental Physiology and Pathophysiology, Laboratory of the Centre for Preclinical Research, Medical University of Warsaw, Warsaw, Poland; [2]Department of Soft Condensed Matter, Institute of Physical Chemistry, Polish Academy of Sciences, Warsaw, Poland; [3]Mass Spectrometry Laboratory, Institute of Biochemistry and Biophysics, Polish Academy of Sciences, Warsaw, Poland; [4]Department of Renal and Body Fluid Physiology, M. Mossakowski Medical Research Centre, Polish Academy of Sciences, Warsaw, Poland

**Abstract** Trimethylamine-oxide (TMAO) is present in seafood which is considered to be beneficial for health. Deep-water animals accumulate TMAO to protect proteins, such as lactate dehydrogenase (LDH), against hydrostatic pressure stress (HPS). We hypothesized that TMAO exerts beneficial effects on the circulatory system and protects cardiac LDH exposed to HPS produced by the contracting heart. Male, Sprague-Dawley and Spontaneously-Hypertensive-Heart-Failure (SHHF) rats were treated orally with either water (control) or TMAO. In vitro, LDH with or without TMAO was exposed to HPS and was evaluated using fluorescence correlation spectroscopy. TMAO-treated rats showed higher diuresis and natriuresis, lower arterial pressure and plasma NT-proBNP. Survival in SHHF-control was 66% vs 100% in SHHF-TMAO. In vitro, exposure of LDH to HPS with or without TMAO did not affect protein structure. In conclusion, TMAO reduced mortality in SHHF, which was associated with diuretic, natriuretic and hypotensive effects. HPS and TMAO did not affect LDH protein structure.

*For correspondence: mufnal@wum.edu.pl

Competing interests: The authors declare that no competing interests exist.

## Introduction

Some clinical studies have shown that increased levels of trimethylamine-oxide (TMAO) in the plasma are associated with an increased risk of adverse cardiovascular events (*Tang et al., 2015*; *Trøseid et al., 2015*; *Tang et al., 2013*). However, other studies have not confirmed this relationship (*Meyer et al., 2016*; *Yin et al., 2015*; *Stubbs et al., 2019*). Furthermore, basic research data regarding the effect of TMAO on the circulatory system are contradictory (*Huc et al., 2018*; *Aldana-Hernández et al., 2019*; *Collins et al., 2016*; *Organ et al., 2016*; *Savi et al., 2018*).

In the plasma, TMAO originates from the liver oxidation of trimethylamine (TMA), a product of gut bacteria metabolism of l-carnitine and choline (*Zeisel and Warrier, 2017*; *Ufnal et al., 2015*). However, another direct source of TMAO in humans is TMAO-rich seafood (*Cheung et al., 2017*; *Yancey and Siebenaller, 2015*). Therefore, populations whose diet are rich in seafood, such as the Japanese, have higher urine TMAO concentrations than those that do not, for example, Americans (*Dumas et al., 2006*; *Holmes et al., 2008*). Interestingly, prevalence and mortality rates of heart

**eLife digest** Heart failure is a common cause of death in industrialized countries with aging populations. Japan, however, has lower rates of heart failure and fewer deaths linked to this disease than the United States or Europe, despite having the highest proportion of elderly people in the world. Dietary differences between these regions may explain the lower rate of heart failure in Japan. The Japanese diet is rich in seafood, which contains nutrients that promote heart health, such as omega-3 fatty acids.

Seafood also contains other compounds, including trimethylamine oxide (TMAO). Fish that live in deep waters undergo high pressures, which can damage their proteins, but TMAO seems to protect the proteins from harm. In humans, eating seafood increases TMAO levels in the blood and urine, but it is unclear what effects this has on heart health. Increased levels of TMAO in the blood are associated with cardiovascular diseases, but scientists are not sure whether TMAO itself harms the heart. A toxic byproduct of gut bacteria called TMA is converted in TMAO in the body, so it is possible that TMA rather than TMAO is to blame.

To assess the effects of dietary TMAO on heart failure, Gawrys-Kopczynska et al. fed the compound to healthy rats and rats with heart failure for one year. TMAO had no effects on the healthy rats. Of the rats with heart failure that were fed TMAO, all of them survived the year, while one third of rats with heart failure that were not fed TMAO died. TMAO-treated rats with heart failure had lower blood pressure and urinated more than untreated rats with the condition.

The experiments suggest that dietary TMAO may mimic the effects of heart failure treatments, which remove excess water and salt and lower pressure on the heart. More studies are needed to confirm whether TMAO has this same effect on humans.

failure (HF) are lower in Japan compared to the US or Europe, despite the fact that Japan has the highest proportion of elderly people in the world (*Nagai et al., 2018*; *Ogawa et al., 2007*).

Marine animals living in deep water, and thus exposed to hydrostatic pressure stress (HPS), accumulate TMAO (*Yancey and Siebenaller, 2015*; *Ma et al., 2014*; *Yancey and Siebenaller, 1999*; *Sarma and Paul, 2013*; *Yin et al., 2017*). Data from biophysical experiments suggest a protective effect of TMAO on cell proteins exposed to HPS. For example, TMAO has been found to stabilize teleost and mammalian lactate dehydrogenase (LDH), a complex tetramer protein that plays an essential role in cellular metabolism (*Yancey and Siebenaller, 1999*).

Notably, in a heart exposed to catecholamine-induced stress or in a hypertensive heart, diastolic-systolic changes in pressure may exceed 220 mmHg. These fluctuations in pressure can happen in humans up to 200 times per minute. These numbers are even higher in small animals. These events may produce an environment in which the hydrostatic pressure changes approximately 100 000 times in 24 hr. However, the effect of HPS produced by the contracting heart on cardiac proteins is obscure.

Recently, we hypothesized that TMAO may benefit the circulatory system by protecting cardiac LDH exposed to HPS produced by diastole-systole-driven changes in the hydrostatic pressure of the contracting heart (*Ufnal and Nowiński, 2019*). Here, we investigated whether a continuous, 12-months-long oral administration of TMAO in healthy Sprague-Dawley rats (SD), in SD exposed to catecholamine stress, and in animal model of heart failure (HF) with reduced ejection fraction (SHHF) exerts beneficial effects on the circulatory system. Furthermore, we evaluated whether TMAO protects the protein structure of cardiac LDH exposed to HPS. To examine the effects of HPS in the context that mimics the environment produced by the contracting heart, we developed a novel experimental system using microfluidics chambers with piezoelectric valves and pressure controllers.

## Results

Spontaneously Hypertensive Heart Failure (SHHF) and age-matched Sprague-Dawley (SD) rats were randomly assigned to either the control group (rats drinking tap water) or the TMAO group (rats drinking TMAO solution in tap water). At the age of 56 weeks, the ISO-control and ISO-TMAO series were given isoprenaline at a dose of 100 mg/kg b.w. to produce catecholamine stress. The experimental protocol is depicted in *Figure 1*.

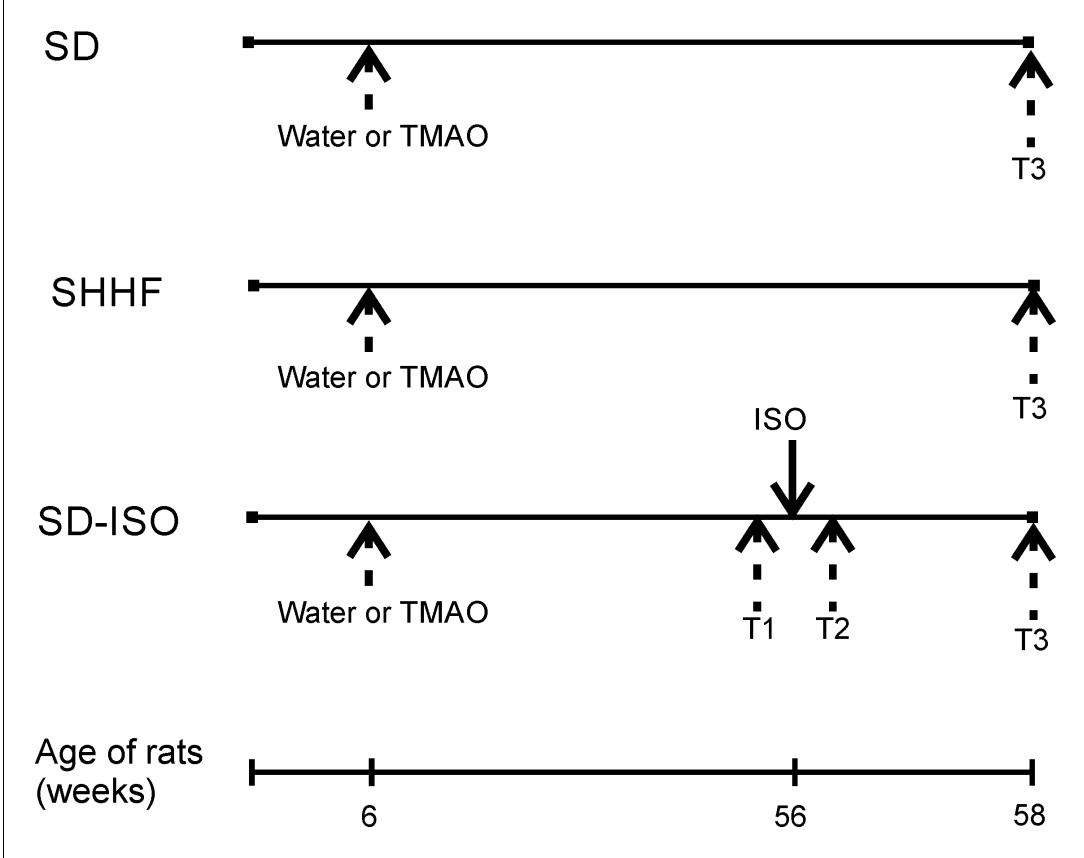

**Figure 1.** Schematic illustration of experimental series. 6-week-old rats started drinking either water (control) or a TMAO solution. SHHF - Spontaneously Hypertensive Heart Failure (SHHF/MccGmiCrl-*Lepr*$^{cp}$/Crl) SHHF, SD – Sprague-Dawley rats, SD-ISO - SD rats treated with ISO at the age of 56 week. ISO - administration of isoprenaline at a dose of 100 mg/kg s.c. T1 - metabolic and echocardiographic measurements, T2 - echocardiographic measurements, T3 - metabolic, echocardiographic and direct hemodynamic measurements.

## SD rats

In general SD rats showed no pathological findings (*Table 1*, *Figures 2* and *3*, and *Figure 3—figure supplement 1*).

### SD rats: Control vs TMAO treatment

#### Survival and water-electrolyte balance

There was no significant difference between the SD-control and SD-TMAO rats in survival rate (100% in both groups), body mass and food intake. The SD-TMAO rats showed 5-times higher plasma TMAO levels than the SD-control group. The SD-TMAO group showed a significantly higher 24 hr urine output and sodium urine excretion. There was no significant difference between the SD-TMAO and SD-control group in plasma Ang II and aldosterone level. However, the SD-TMAO rats showed higher vasopressin concentration (*Table 1*).

#### Circulatory parameters

There was no significant difference between the SD-control and SD-TMAO group in hemodynamic parameters measured directly and with echocardiography, however, the SD-TMAO rats showed higher SV and HR (*Table 1*).

#### Histopathology

There were no pathological changes in the heart, lungs and kidneys in either the SD-control or SD-TMAO group (*Figure 3* and *Figure 3—figure supplement 1*).

### Gene expression

The SD-TMAO rats showed significantly lower expression of AT1 receptors (AT1R) in the heart. In the kidneys, SD-TMAO rats showed significantly higher expression of renin but lower expression of AT2 receptors (AT2R) and a trend towards a lower expression of AT1R (*Figures 4* and *5*).

## SHHF rats

In general, the SHHF animals showed characteristics of hypertrophic cardiomyopathy with compromised systolic function including substantially increased heart mass and plasma NT-proBNP, decreased stroke volume and ejection fraction as well as lung edema (*Table 2*, *Figure 2*). Histological evaluation of the heart revealed dilated cardiomyopathy that is a moderate increase in the diameter of cardiomyocytes, enlargement of the nucleus and a reduction of cytoplasmic acidity (*Figure 6*). In the lungs, a passive hyperemia with thickening of the interalveolar septa, a weak focal parenchymal edema and a moderate stromal connective tissue hyperplasia were present. There were no significant pathological changes in the kidneys (*Figure 6*).

### SHHF rats: Control vs TMAO treatment
### Survival and water-electrolyte balance

All SHHF-TMAO rats (n = 9) survived from the beginning of the experiment till the age of 58 weeks, that is the time of anesthesia before echocardiography. In contrast, three out of nine animals (33%) in the SHHF-control group died. Specifically, two rats were euthanized due to hemiparalysis (ischemic stroke) and dyspnea (post-mortem lung edema), and one died spontaneously (post-mortem lung edema), at the age of 52, 56 and 57 weeks, respectively. The log-rank test showed a trend (p=0.0651) towards higher survival in the SHHF-TMAO than in SHHF-control group (*Figure 7*). One SHHF-TMAO animal died during anesthesia before echocardiographic examination. The following numbers in each group were included for further analysis: SHHF-TMAO (n = 8) and SHHF-control (n = 6).

There was no significant difference between the groups in food intake. The SHHF-TMAO animals showed 3–4-fold higher plasma TMAO level than the SHHF-control rats. The SHHF-TMAO rats had a significantly higher 24 hr urine output and sodium excretion than the SHHF-control rats. The SHHF-TMAO group showed significantly lower plasma sodium and Ang II levels, but there were no differences in aldosterone and vasopressin plasma levels (*Table 2*).

### Circulatory parameters

The SHHF-control and SHHF-TMAO group showed hypertrophic cardiomyopathy with compromised systolic functions and increased plasma NT-proBNP (*Figures 2* and *6*, *Table 2*). The SHHF-TMAO rats had a significantly lower diastolic blood pressure and a trend towards lower plasma NT-proBNP (p=0.09), higher stroke volume (p=0.053) and higher ejection fraction (p=0.06), (*Table 2*).

### Histopathology

The general histological picture of myocardium and cardiomyocyte morphology did not differ significantly between the groups. Morphometric analysis did not show significant differences in the degree of myocardial fibrosis between the groups. However, in the TMAO group there was a trend towards less myocardial fibrosis (p=0.07), cardiomyocyte bands were more visible, and smaller inflammatory infiltration was found. The histological picture of the lungs and kidneys did not differ significantly between SHHF-control and SHHF-TMAO groups (*Figure 6*).

### Gene expression

The SHHF-TMAO rats had a significantly lower expression of AT1R and a significantly higher expression of AT2R in the heart. In the kidneys, the SHHF-TMAO rats showed a significantly higher expression of renin and lower expression of AT1AR (*Figures 4* and *5*).

## SD-ISO rats

In general, the SD animals subjected to catecholamine stress (isoprenaline) showed some characteristics of Takotsubo-like cardiomyopathy, including a mild degree of apical akinesis/dyskinesis, edema of cardiomyocytes, increased NT-proBNP level and mild lung edema (*Figures 2* and *8*, *Table 3*).

Numerous, scattered foci of banded mononuclear cell infiltration were present in the myocardium. Severe hyperemia of myocardial capillaries and arterioles and small organized foci of myocardial extravasation were present. Nevertheless, the majority of the myocardium remained normal in structure. The lungs presented transudate in the alveoli. In the kidneys, there was a weak congestion in the medulla and renal bodies. A small number of tubules filled with an acidic substance was present. Stimulation of stromal fibrocytes without production of connective tissue fibers was observed (*Figure 8*).

### SD-ISO rats: Control vs TMAO treatment
### Survival and water-electrolyte balance

There was no significant difference between ISO-control and ISO-TMAO groups in survival rate (9/10 vs 10/10, respectively). The ISO-TMAO rats showed five times higher TMAO plasma level than the ISO-control rats. There was no significant difference in food and water intake between the groups. There was also no significant difference between the groups in 24 hr urine output, however, the ISO-TMAO rats tended to have higher natriuresis (*Table 3*).

### Circulatory parameters

The ISO-TMAO rats showed significantly lower systolic and diastolic blood pressure, significantly lower plasma NT-proBNP and lower LV ESV and LVEDP (*Table 3*).

### Histopathology

The histological picture of the heart, lungs and kidneys did not differ significantly between the groups (*Figure 8*).

### Gene expression

In the heart, the ISO-TMAO rats showed significantly lower expression of ATG, AT1R and AT2R. In the kidneys, the ISO-TMAO rats showed significantly lower expression of ATG and AT1R and significantly higher expression of renin (*Figures 4* and *5*).

## Effect of TMAO on renal excretion in SD rats - acute studies

We evaluated changes in renal excretion induced by TMAO, urea and saline intravenous administration in acute experiments. Results are summarized in *Figure 9*. Only TMAO induced diuresis. The pattern of diuresis and total solutes excretion induced by TMAO were similar. Increases of V and UosmV induced by TMAO were associated with transient decrease of Uosm, whereas UNaV and UKV were not affected. This indicates that TMAO did not affected the tubular transport of sodium and potassium but induced osmotic diuresis. The bolus infusions of TMAO or saline produced a transient increase in ABP with no changes in HR, which was followed by a decrease in ABP below the baseline (by $6 \pm 3$ mmHg). There was no significant correlation between changes in ABP and diuresis.

## Effect of TMAO on structure of LDH exposed to HPS and increased temperature

Labeled LDH was stable in PBS solution, showing no tendency for spontaneous aggregation. The value of diffusion coefficients measured by FCS at 25°C was $49.2 \pm 3.3$ µm2/s. This corresponds to a hydrodynamic radius of around 5.0 nm, which is in line with previously reported values (*Zipper and Durchschlag, 1998*).

The tertiary and quaternary structures of LDH, with and without TMAO, were not influenced by a 24 hr treatment with HPS (pressure oscillations mimicking those of a rat heart), (*Figure 10A*). Tests performed using a different pressure oscillation system, where pressures up to 1000 mmHg were applied, did not detect observable changes in the protein structure (see *Figure 10—figure supplement 1*).

The incubation of LDH at atmospheric pressure and elevated temperatures (50–80°C) changed the diffusion coefficient of LDH, indicating the dissociation of LDH tetramers, as well as protein denaturation and aggregation. The addition of 1M TMAO produced a moderate stabilizing effect on LDH, shifting the threshold of observed protein morphology change towards higher temperatures

**Table 1.** Metabolic, renal and cardiovascular parameters in 58-week-old normotensive SD rats. Sprague-Dawley rats maintained on either water (SD-control, n = 6–10) or TMAO solution (SD-TMAO, n = 7–10). Creatinine clearance calculated as urine creatinine x urine output (ml/min)/plasma creatinine. LVEDV - left ventricle end diastolic volume, LVESV - left ventricle end systolic volume, SV – stroke volume, EF - ejection fraction, IVSs(d) - intraventricular septum diameter during systole and diastole, respectively, LVEDP - pressure in the left ventricle during the end of diastole measured directly with a catheter, +dP/dt - maximal slope of systolic ventricular pressure increment, - dP/dt - maximal slope of diastolic ventricular pressure decrement. Values are means,± SD. P values by t-test or Mann-Whitney U test.

| Group/Parameter | SD-control | SD-TMAO | P |
|---|---|---|---|
| **Survival, energy and water balance** | | | |
| Survival from the study onset (%, n) | 100% (10/10) | 100% (10/10) | - |
| Body mass (g) | 446.4 ± 40.63 | 452.86 ± 37.11 | 0.36 |
| 24 hr food intake (g) | 19.99 ± 2.83 | 21.49 ± 2.64 | 0.12 |
| 24 hr water intake (g) | 31.56 ± 10.02 | 36.66 ± 5.18 | 0.09 |
| 24 hr urine output (g) | 18.66 ± 2.35 | 22.66 ± 6.49 | **0.04** |
| Tibia length (cm) | 4.31 ± 0.1 | 4.25 ± 0.15 | 0.15 |
| **TMAO** | | | |
| Plasma TMAO (μmol/L) | 6.55 ± 0.65 | 39.73 ± 20.6 | **<0.001** |
| 24 hr TMAO urine excretion (μmoles) | 5.96 ± 1.49 | 103.05 ± 56.7 | **<0.001** |
| **Heart mass** | | | |
| Heart mass (g) | 1.44 ± 0.08 | 1.46 ± 0.14 | 0.38 |
| **Arterial blood pressure and heart rate** | | | |
| Systolic (mmHg) | 129.72 ± 8.56 | 127.07 ± 5.84 | 0.87 |
| Diastolic (mmHg) | 80.56 ± 13.3 | 86.61 ± 9.74 | 0.15 |
| HR (beats/min) | 333.9 ± 45 | 364.4 ± 8.6 | **0.04** |
| **Echocardiographic parameters** | | | |
| LVEDV (mL) | 0.47 ± 0.15 | 0.57 ± 0.1 | 0.06 |
| LVESV (mL) | 0.12 ± 0.04 | 0.13 ± 0.03 | 0.44 |
| IVSs (cm) | 0.35 ± 0.03 | 0.35 ± 0.03 | 0.42 |
| IVSd (cm) | 0.24 ± 0.03 | 0.25 ± 0.03 | 0.19 |
| SV (mL) | 0.36 ± 0.11 | 0.44 ± 0.1 | **0.04** |
| EF (%) | 75.63 ± 3.02 | 77.13 ± 6.01 | 0.27 |
| **Left ventricle hemodynamic parameters (direct measurements)** | | | |
| LVEDP (mmHg) | 4.12 ± 0.78 | 4.25 ± 0.92 | 0.45 |
| dP/dt (mmHg/ms) | 6.54 ± 1.02 | 7.31 ± 1.23 | 0.12 |
| -dP/dt (mmHg/ms) | 5.00 ± 0.69 | 5.26 ± 0.45 | 0.21 |
| **Plasma NT-proBNP** | | | |
| NT-proBNP (pg/mL) | 24.79 ± 8.1 | 18.61 ± 8.17 | 0.22 |
| **Electrolyte balance** | | | |
| Serum sodium (mmol/L) | 138.86 ± 2.27 | 138.44 ± 0.50 | 1.0 |
| 24 hr sodium urine excretion (mmoles) | 1.76 ± 0.23 | 2.11 ± 0.08 | **0.003** |
| Serum potassium (mmol/L) | 5.27 ± 0.89 | 5.13 ± 0.17 | 0.71 |
| 24 hr potassium urine excretion (mmoles) | 2.83 ± 0.58 | 3.00 ± 0.12 | 0.23 |
| Serum creatinine clearance (mL/min) | 1.15 ± 0.18 | 1.16 ± 0.07 | 0.67 |
| **Hormones** | | | |
| Angiotensin II (pg/mL) | 244.93 ± 35.55 | 250.07 ± 64.95 | 0.43 |
| Aldosterone (pg/mL) | 897.05 ± 95.34 | 925.61 ± 75.29 | 0.27 |
| Vasopressin (ng/mL) | 0.92 ± 0.98 | 1.78 ± 0.59 | **0.02** |

The online version of this article includes the following source data for Table 1:

**Source data 1.** Metabolic, renal and cardiovascular parameters in 58-week-old normotensiveSD rats.

(**Figure 10B**). Specifically, it seems that the gradual dissociation of tetramers to monomers occurred above 55°C, which was followed by the denaturation of the tertiary protein structure at higher temperatures. At a concentration of LDH >3 nM, the aggregation of monomers prevailed. At a lower concentration, the aggregation progressed more slowly, if at all, while any aggregates that did form, were too sparse to influence the measurement results. Further work, including probing a broader matrix of LDH concentrations and temperatures, is needed to confirm these initial findings. Nevertheless, in all the experiments, presence of TMAO shifted the threshold of change in LDH morphology towards higher temperatures and diminished the magnitude of the change. This suggests a stabilizing effect of TMAO on the native structure of the protein.

## Discussion

Our study provides evidence that TMAO exerts a beneficial effect in heart failure rats. The beneficial effect of TMAO is associated with diuretic, natriuretic and hypotensive actions rather than a stabilizing effect on cardiac LDH. Specifically, we found that HPS at the magnitude substantially greater than that generated by a contracting heart did not affect cardiac LDH protein structure with or without TMAO.

Plasma TMAO originates from trimethylamine (TMA), a product of gut bacteria, which is oxidized by the liver to form TMAO. It can also be obtained from dietary TMAO-rich seafood (**Zeisel and Warrier, 2017**; **Ufnal et al., 2015**). Several years ago, Tang and collaborators showed that increased plasma TMAO is associated with an increased risk of major adverse cardiovascular events (**Tang et al., 2013**), which generated numerous clinical and experimental studies, suggesting negative effects of TMAO (for review, [**Zeisel and Warrier, 2017**; **Onyszkiewicz et al., 2020**]). However, other clinical studies did not find an association between high plasma TMAO and increased cardiovascular risk (**Meyer et al., 2016**; **Yin et al., 2015**; **Stubbs et al., 2019**).

Our previous studies suggest that TMAO is only a surrogate marker and that the toxic cardiovascular effects are caused not by TMAO itself, but by a TMAO precursor, that is TMA (**Jaworska et al., 2019a**; **Jaworska et al., 2019b**; **Jaworska et al., 2019c**). Notably, the latter is a well-established toxin (**Pospischil et al., 2017**). Since TMA is oxidized (likely, detoxified) by the liver to TMAO, the increased plasma TMAO is simply a proxy to the plasma levels of the toxic TMA. What is more, recent experimental studies provide evidence for a beneficial action of TMAO at low doses (**Huc et al., 2018**; **Aldana-Hernández et al., 2019**; **Collins et al., 2016**).

Here, we investigated the effect of TMAO in (i) healthy SD rats, (ii) SHHF (SHHF/MccGmiCrl-Lepr$^{cp}$/Crl) which showed characteristics of heart failure with compromised systolic function and (iii) SD subjected to catecholamine stress (isoprenaline).

Our findings show that increased dietary TMAO, which elevates plasma TMAO level 3–5-fold, increased diuresis and natriuresis in SD and SHHF rats. This is associated with reduced mortality and favorable hemodynamic and biochemical changes in HF rats.

In this study, during a stress-free 52 week observation, the fatality rate of the SHHF animals treated with TMAO was 0% in comparison to 33% in the untreated group. This was associated with several beneficial hemodynamic and biochemical characteristics in the TMAO-treated SHHF rats, namely; a lower diastolic BP, higher diuresis and natriuresis, a lower expression of AT1R in the heart with concomitant increase in AT2R, and a trend towards lower cardiac fibrosis. Regarding the latter, the tissue angiotensin system contributes significantly to remodeling of the heart, and its inhibition is critical in the treatment of heart failure (**Leong et al., 2019**).

Some of the characteristic features of HF are inflammation and increased sympathetic activity. In this regard, the SHHF-TMAO group had a significantly increased plasma level of IL-10, an anti-inflammatory cytokine, and a trend towards lower plasma TNF-α, a pro-inflammatory mediator. Significantly lower diastolic blood pressure together with mildly lower heart rate may also suggest lower sympathetic activity in the SHHF-TMAO group. However, to fully address this issue further studies evaluating concentration of catecholamines metabolites in urine are needed. It is worth stressing that due to the high mortality in the SHHF-control group, the final analysis of hemodynamic and

biochemical parameters included only the survivors that is the healthiest rats. It is interesting to speculate, had all the SHHF-control rats survived to the experimental endpoints, whether the differences between the SHHF-TMAO and SHHF-control would have been even greater.

Finally, a key characteristic of HF is fluid retention. We think that the beneficial effects of TMAO described above were secondary to the TMAO-produced increase in diuresis and natriuresis, which is a cornerstones of HF treatment.

We assumed that the diuretic and natriuretic effects of TMAO resulted from osmotic diuresis. To evaluate the notion, we performed additional acute experiments in rats. We found that intravenous administration of TMAO, but not saline given at the same volume, significantly increased diuresis in anesthetized rats. Furthermore, in chronic studies, TMAO-treated rats showed increased expression of renin in the kidneys. This is a characteristic response to osmotic diuresis, associated with a decreased concentration of sodium in filtrate reaching distal convoluted tubule. Increased diuresis in TMAO-treated rats was present despite elevated plasma levels of vasopressin which allows the reabsorption of water from the filtrate in collecting ducts. The reabsorption is driven by the osmotic pressure gradient between the filtrate and the kidney medulla. In TMAO-treated rats, the osmotic gradient was likely decreased due to the high osmotic activity of TMAO and high TMAO levels in the filtrate. This could decrease the reabsorption of water from the filtrate despite the vasopressin-induced increase in the permeability of collecting ducts to water.

To evaluate diuretic effect of TMAO, we compared TMAO to urea, which was used as a diuretic treatment of advanced HF before being replaced by thiazide diuretics (*Shanoff, 1970*; *Crawford, 1925*). Notably, both molecules are nitrogenous compounds, have a similar molecular weight, and play the role of an osmolyte in animals (*Yancey and Siebenaller, 2015*). In our experiments, TMAO produced significant diuresis whereas urea did not. This was likely due the fact that we evaluated the same (equimolar) doses of urea and TMAO, whereas physiological concentrations of TMAO in the plasma is radically lower (micromoles/L) in comparison to urea (millimoles/L). Therefore, our findings suggest that TMAO exerts a significantly more potent diuretic effect than urea.

A number of biological and biophysical studies indicate that TMAO is not only an osmolyte but also a piezolyte, that is stabilizes structural proteins and enzymes, such as LDH, in conditions of increased hydrostatic pressure (*Yancey and Siebenaller, 1999*; *Sarma and Paul, 2013*; *Yin et al., 2017*). Therefore, the accumulation of TMAO in the bodies of deep-water animals may protect cell proteins from HPS (*Yancey and Siebenaller, 2015*; *Yancey and Siebenaller, 1999*).

The effect of HPS produced by the contracting heart on cardiac proteins is unclear. Based on studies showing that TMAO stabilizes teleost and mammalian lactate dehydrogenases against inactivation by hydrostatic pressure (*Yancey and Siebenaller, 1999*; *Al-Ayoubi et al., 2019*; *Yancey et al., 2001*), we hypothesized that TMAO may exert a protective effect on the circulatory system by stabilizing the protein structure of cardiac LDH, a complex enzyme that plays essential role in cardiac metabolism (*Ufnal and Nowiński, 2019*).

To evaluate this hypothesis, we built a unique experimental setup to mimic the changes in hydrostatic pressure that occur in the heart under conditions of catecholamine stress. These in vitro experiments showed that changes in hydrostatic pressure generated by the contracting heart, and even much higher pressures, do not disturb the protein structure of cardiac LDH. Finally, the addition of TMAO did not produce any effect, suggesting neither positive nor negative effect of TMAO on LDH exposed to HPS. Nevertheless, we showed that TMAO stabilized LDH exposed to other denaturant, that is high temperature, which is in line with other studies (*Schummel et al., 2018*). In general, changes in protein structure lead to perturbed (or even the loss of) activity of proteins. However, further studies are needed to evaluate if LDH activity would be affected by HPS, TMAO and temperature in a similar manner as LDH protein structure.

Altogether, our study shows that TMAO exerts beneficial effects in cardiovascular pathologies associated with fluid retention such as HF. The beneficial effects of TMAO appear to stem from its diuretic action rather than from the protein-stabilizing effect of TMAO on LDH, which was described for deep-sea animals, but was found not be involved here. It may be that the hydrostatic pressure generated by a contracting heart is far lower than the deep-sea pressures and is kinetic (pulsing and moving) rather than static. In this regard, there is evidence that static pressure may have very different effects on cells and proteins than kinetic pressure (*Kaarniranta et al., 1998*). Nevertheless, the stabilizing effect of TMAO on other, less stable, cardiac proteins exposed to HPS cannot be excluded.

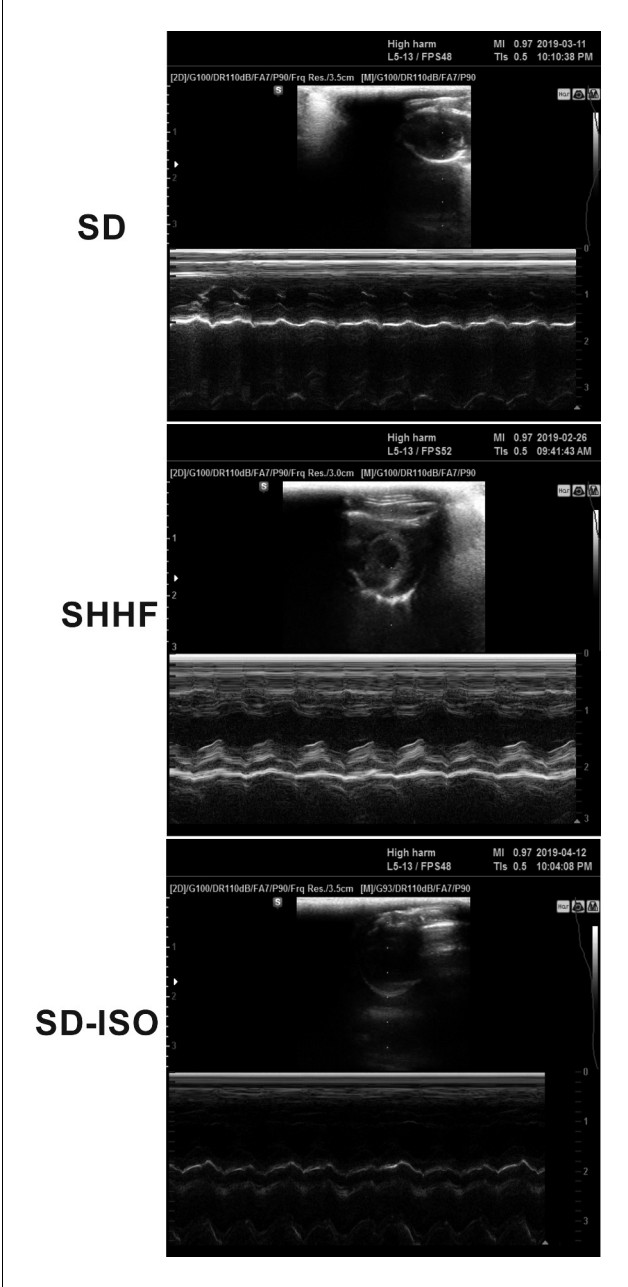

**Figure 2.** Echocardiography in 58-wk-old rats. SHHF - Spontaneously Hypertensive Heart Failure (SHHF/MccGmiCrl-*Lepr^cp*/Crl), SD – Sprague-Dawley rats, SD-ISO: SD rats treated with ISO at the age of 56 week. ISO: administration of isoprenaline at a dose of 100 mg/kg s.c. SD: Left ventricular systolic function is normal/preserved. Left and right ventricular diameter is normal. Left ventricular wall thickness is normal. Left atrial diameter is normal. SHHF: Septal hypokinesis. Left ventricular free wall is hypertrophic. Endocardium is hyperechogenic. SD-ISO: Septal hypokinesis. Left ventricular end-systolic diameter is increased. Left atrial enlargement.

## Limitations

A limitation of our study is that biochemical and hemodynamic measurements were performed only at the end of the treatment. This is because we aimed to avoid stress-related circulatory complications in SHHF rats, which are very prone to lethal cardiovascular events.

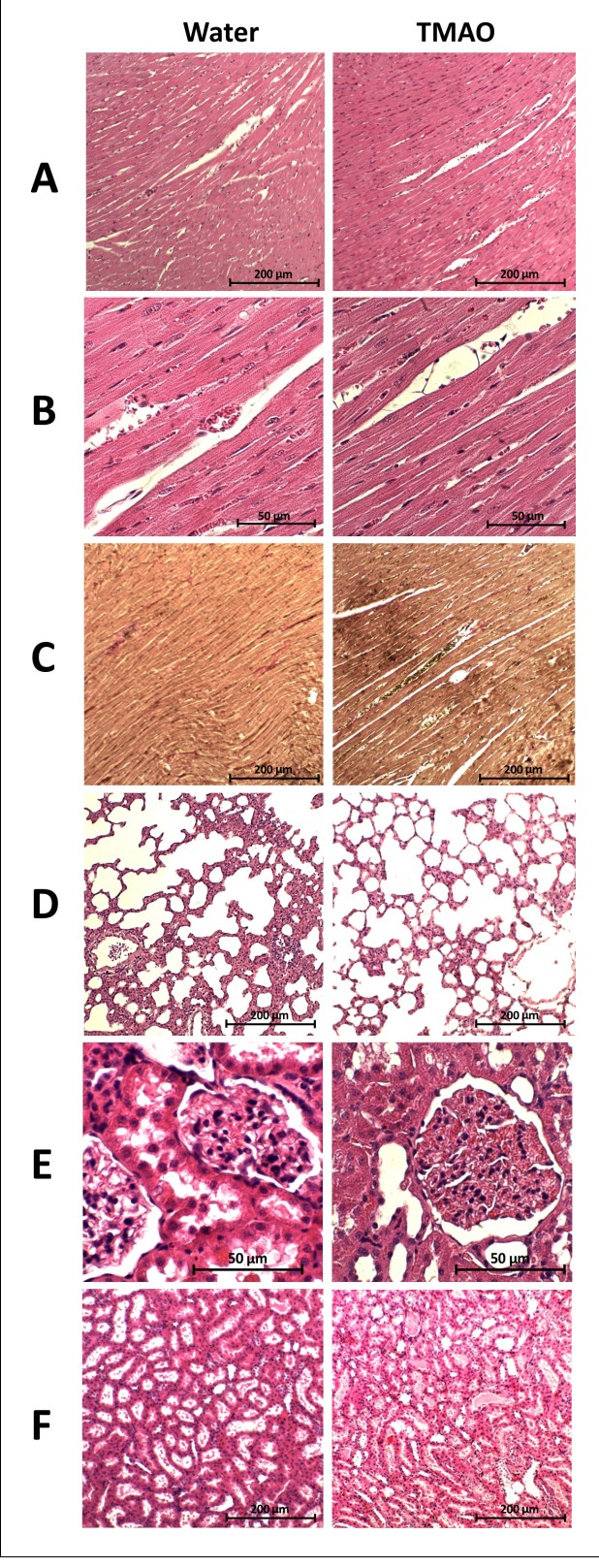

**Figure 3.** Histopathological picture of the heart, lungs and kidneys in the Sprague-Dawley rats drinking either water (control group) or TMAO solution.  A - myocardium; hematoxylin-eosin (HE) staining at magnification x10; B - myocardium; HE staining at magnification x40; C - myocardium; van Gieson staining at magnification x10; D - *Figure 3 continued on next page*

*Figure 3 continued*

lungs; HE staining at magnification x10; E – kidney - renal cortex, renal bodies; HE staining at magnification x40; F - kidney - renal medulla; hematoxylin-eosin staining at magnification x10.

The online version of this article includes the following figure supplement(s) for figure 3:

**Figure supplement 1.** Comparison of the histopathological picture between SD, SHHF and ISO-SD.

## Conclusions

TMAO, a molecule present in seafood and a derivate of gut bacteria metabolism, exerts beneficial effects in HF rats. These benefits might be derived from the diuretic, natriuretic and hypotensive properties of TMAO. The hydrostatic pressure stress generated by the contracting heart does not affect LDH protein structure. Further studies designed to evaluate TMAO-dependent diuretic and natriuretic effects are needed, as TMAO may serve as a naturally occurring diuretic agent in diseases associated with fluid retention e.g. heart failure.

# Materials and methods

### Key resources table

| Reagent type (species) or resource | Designation | Source or reference | Identifiers | Additional information |
|---|---|---|---|---|
| Strain, strain background (Rattus norvegicus, male) | SHHF/MccGmi Crl-*Lepr^cp*/Crl | Charles River Laboratories (USA) | RRID:RGD_2313221 | |
| Commercial assay or kit | NT-proBNP | FineTest | cat. no. ER0309 | |
| Commercial assay or kit | aldosterone | Cayman Chemicals | cat. no. 501090 | |
| Commercial assay or kit | vasopressin | Biorbyt | cat. no. orb410987 | |
| Commercial assay or kit | angiotensin II | FineTest | cat. no. ER1637 | |
| Commercial assay or kit | TNFα | R and D System | cat. no. RTA00 | |
| Commercial assay or kit | IL-10 | R and D System | cat. no. R1000 | |
| Commercial assay or kit | angiotensinogen | Bio-Rad | Unique Assay ID: qRnoCED0051666 | |
| Commercial assay or kit | angiotensin II receptor type 1a | Bio-Rad | Unique Assay ID: qRnoCID0052626 | |
| Commercial assay or kit | angiotensin II receptor type 1b | Bio-Rad | Unique Assay ID: qRnoCED0005729 | |
| Commercial assay or kit | angiotensin II receptor type 2 | Bio-Rad | Unique Assay ID: qRnoCED0007551 | |
| Commercial assay or kit | transforming growth factor-beta | Bio-Rad | Unique Assay ID: qRnoCED0007638 | |
| Commercial assay or kit | renin | Bio-Rad | Unique Assay ID: qRnoCID0008721 | |
| Commercial assay or kit | metalloproteinase inhibitor 2 | Bio-Rad | Unique Assay ID: qRnoCID0001559 | |
| Commercial assay or kit | Beta-actin | Bio-Rad | Unique Assay ID: qRnoCED0018219 | |
| Commercial assay or kit | TMAO | abcr GmbH | cat. no. AB 109058 | |
| Commercial assay or kit | isoprenaline hydrochloride | Sigma-Aldrich | cat. no. I5627 | |

*Continued on next page*

*Continued*

| Reagent type (species) or resource | Designation | Source or reference | Identifiers | Additional information |
|---|---|---|---|---|
| Software, algorithm | CFX Manager | Bio-Rad | RRID:SCR_017251 | |
| Software, algorithm | SymPhoTime 64 | PicoQuant | RRID:SCR_016263 | |
| Software, algorithm | AcqKnowledge Software | Biopac Systems, Inc | RRID:SCR_014279 | |

## Animals

The study was performed according to Directive 2010/63 EU on the protection of animals used for scientific purposes and approved by the Local Bioethical Committee in Warsaw (permission:100/ 2016 and 098/2019). 4–5 week-old, male, lean Spontaneously Hypertensive Heart Failure (SHHF/ MccGmiCrl-*Lepr*$^{cp}$/Crl) rats were purchased from Charles River Laboratories (USA). Age-matched Sprague-Dawley (SD) rats were obtained from the Central Laboratory for Experimental Animals, Medical University of Warsaw, Poland.

## Study protocol

Six-week-old SHHF (n = 18) and SD (n = 40) were randomly assigned to either control groups (rats drinking tap water) or the TMAO groups (rats drinking TMAO solution in tap water, TMAO - abcr GmbH - Karlsruhe, Germany, 333 mg/l). While no specific randomization method was used, rats from one cage were assigned to different groups. The dose of TMAO was selected in order to increase the plasma TMAO concentration by 3–5 times (to mimic possible physiological concentrations) and to avoid suprapharmacological effects of TMAO, based on our previous study (*Huc et al., 2018*).

Rats were housed in groups of 2–3 animals, in polypropylene cages with environmental enrichment, 12 hr light/12 hr dark cycle, temperature 22–23oC, humidity 45–55%, fed standard laboratory diet (0.19% Na, Labofeed B standard, Kcynia, Poland) and water ad libitum.

SHHF-TMAO (n = 9), SHFF-control (n = 9), SD-TMAO (n = 10), SD-control (n = 10) were not subjected to any interventions except standard animal care until the age of 58 weeks. At the age of 56 weeks the ISO-control (n = 10) and ISO-TMAO (n = 10) series were given (s.c.) isoprenaline at a dose of 100 mg/kg b.w. (isoprenaline hydrochloride, Sigma-Aldrich, Saint Louis, MO, USA) to produce catecholamine stress as previously described (*Sachdeva et al., 2014*). The experimental protocol is depicted in *Figure 1*.

## Experimental protocol in SD and SHHF

58-week-old rats were maintained in metabolic cages for 2 days to evaluate the 24 hr water and food balance and to collect urine for analysis. The next day, the rats underwent an echocardiogram using a Samsung HM70: an ultrasound system equipped with a linear probe 5–13. MHz. After the echo examination the rats were anaesthetized with urethane (1.5 g/kg b.w. i.p., Sigma-Aldrich) and the left femoral artery was cannulated with a polyurethane catheter for arterial blood pressure (ABP) recordings. The recordings were started 40 min after the induction of anesthesia and 15 min after inserting the arterial catheter. After 10 min of ABP recordings, a Millar Mikro-Tip SPR-320 (2F) pressure catheter was inserted via the right common carotid artery and simultaneous left ventricle pressure (LVP) and ABP recordings were performed. The catheter was connected to a Millar Transducer PCU-2000 Dual Channel Pressure Control Unit (Millar, USA) and Biopac MP 150 (Biopac Systems, USA). After hemodynamic recordings, blood from the right ventricle of the heart was taken and rats were euthanized by decapitation. The heart, the lungs and the kidneys were harvested for histological and molecular analysis.

## Experimental protocol in SD-ISO

56-week-old rats were housed in metabolic cages for 2 days to evaluate the 24 hr water and food balance and to collect urine for analysis. Echocardiography was performed as described above. The next day, rats were given isoprenaline (100 mg/kg, s.c.). 24 hr after the administration of ISO, the

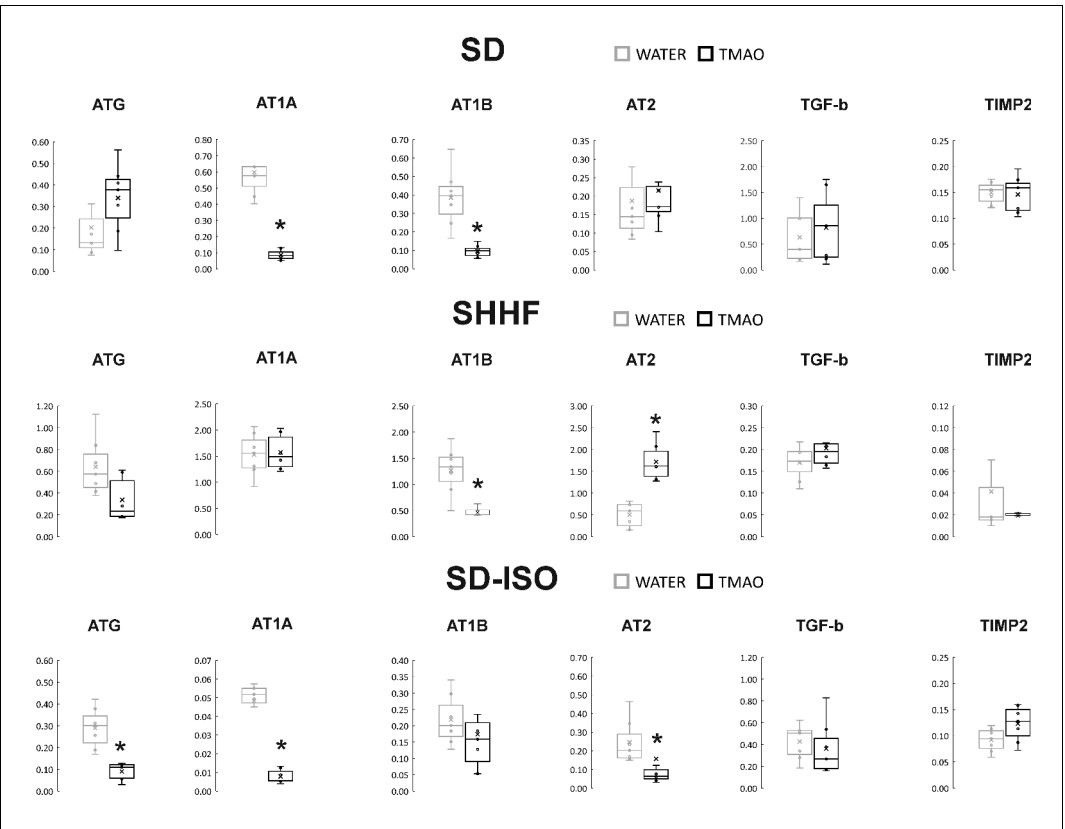

**Figure 4.** Real-time RT-PCR analysis, heart. Box plot comparing the expression profiles of ATG (angiotensinogen), AT1A (angiotensin II receptor type 1a), AT1B (angiotensin II receptor type 1b), AT2 (angiotensin II receptor type 2), TGF-b (transforming growth factor-beta), TIMP2 (metalloproteinase inhibitor 2) in hearts of SD – Sprague-Dawley rats, SHHF - Spontaneously Hypertensive Heart Failure (SHHF/MccGmiCrl-$Lepr^{cp}$/Crl), SD-ISO - SD rats treated with ISO at the age of 56 week, drinking either water (control groups) or TMAO solution, (n = 6–7 for each series). × - mean value, * indicates significant difference compared with the control group. *p<0.05 by t-test or Mann-Whitney U test.

The online version of this article includes the following source data for figure 4:

**Source data 1.** Real-time RT-PCR analysis, expression profiles in heart.

echocardiogram was repeated. Eight days after the ISO-treatment, the 24 hr food and water intake was evaluated and an echocardiogram was performed. Afterwards, the rats were anaesthetized with urethane (1.5 g/kg b.w. i.p., Sigma-Aldrich, Poland) and the hemodynamic measurements were taken, including ABP and LVP recordings as described above for SHHF and SD rats.

## TMAO and general biochemistry evaluation

Plasma and urine concentrations of TMAO were measured using Waters Acquity Ultra Performance Liquid Chromatograph coupled with a Waters TQ-S triple-quadrupole mass spectrometer. The mass spectrometer was operated in the multiple-reaction monitoring (MRM) - positive electrospray ionization (ESI) mode, as previously described (*Jaworska et al., 2017*).

Serum and urine sodium, potassium, creatinine and urea were analyzed using a Cobas 6000 analyzer (Roche Diagnostics, Indianapolis, USA).

## ELISA test

The following ELISA kits were used for the evaluation: NT-proBNP (FineTest, cat. no. ER0309), aldosterone (Cayman Chemicals, cat. no. 501090), vasopressin (Biorbyt, cat. no. orb410987), angiotensin II (FineTest, cat. no. ER1637), TNFα and IL-10 (cat. no. RTA00 and R1000, respectively, R and D System). All procedures were carried out according to the standard protocol supplied with the ELISA

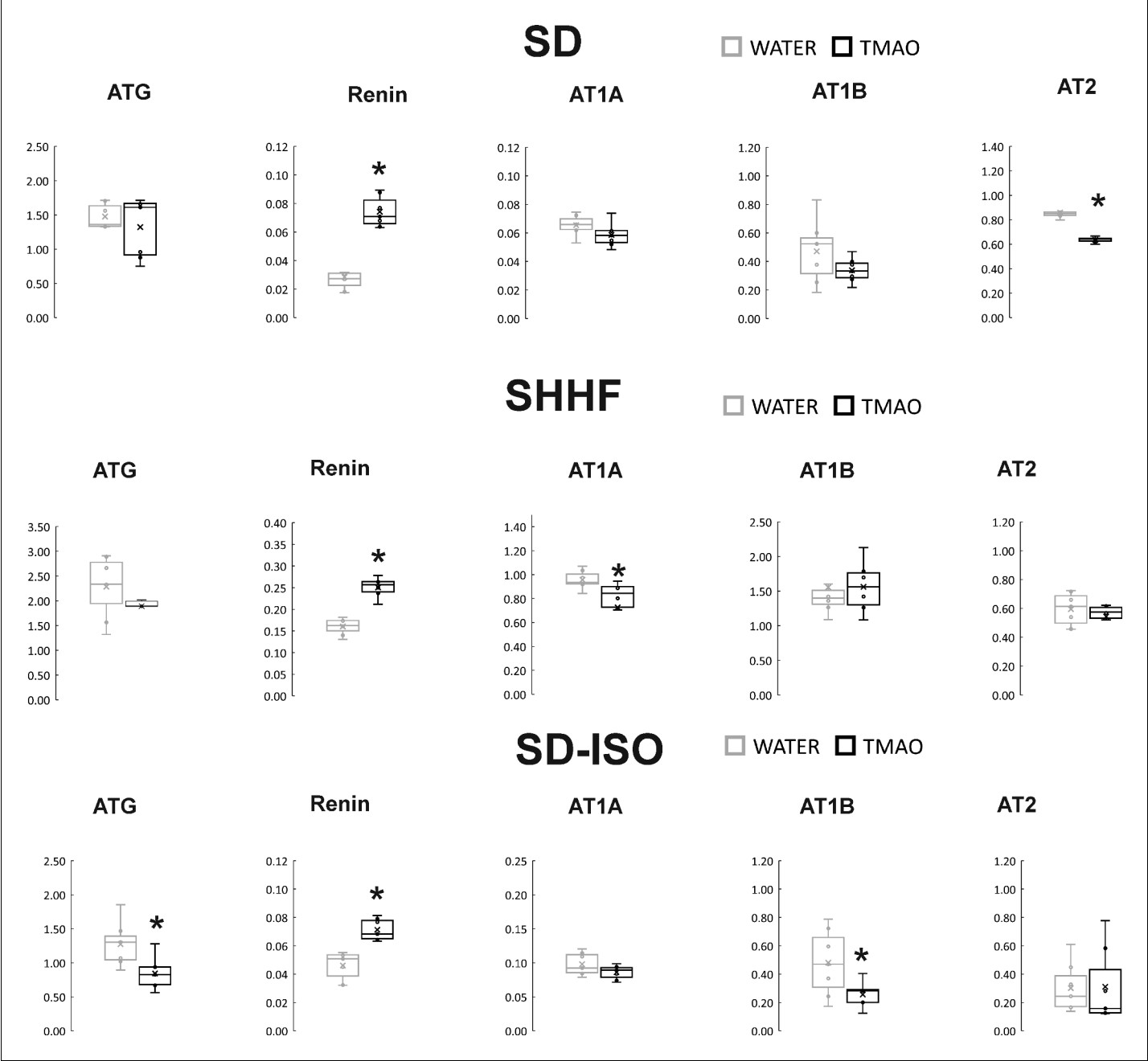

**Figure 5.** Real-time RT-PCR analysis, kidneys. Box plot comparing the expression profiles of ATG (angiotensinogen), AT1A (angiotensin II receptor type 1a), AT1B (angiotensin II receptor type 1b), AT2 (angiotensin II receptor type 2), renin in kidneys of SD – Sprague-Dawley rats, SHHF - Spontaneously Hypertensive Heart Failure (SHHF/MccGmiCrl-*Lepr^cp*/Crl), SD-ISO - SD rats treated with ISO at the age of 56 week, drinking either water (control groups) or TMAO solution, (n = 6–7 for each series). × - mean value,* indicates significant difference compared with the control group. *p<0.05 by t-test or Mann-Whitney U test.

The online version of this article includes the following source data for figure 5:

**Source data 1.** Real-time RT-PCR analysis, expression profiles in kidneys.

**Table 2.** Metabolic, renal and cardiovascular parameters in 58-week-old SHHF rats.

Spontaneously Hypertensive Heart Failure (SHHF/MccGmiCrl-*Lepr*[cp]/Crl) rats maintained on either water (SHHF-control, n = 5–6) or TMAO solution (SHHF-TMAO, n = 7–9).Creatinine clearance calculated as urine creatinine x urine output (ml/min)/plasma creatinine. LVEDV - left ventricle end diastolic volume, LVESV - left ventricle end systolic volume, SV – stroke volume, EF - ejection fraction, IVSs (d) - intraventricular septum diameter during systole and diastole, respectively, LVEDP - pressure in the left ventricle during the end of diastole measured directly with a catheter, +dP/dt - maximal slope of systolic ventricular pressure increment, - dP/dt - maximal slope of diastolic ventricular pressure decrement. Values are means,± SD. P values by t-test or Mann-Whitney U test.

| Group/ Parameter | SHHF-control | SHHF-TMAO | P |
|---|---|---|---|
| **Survival, Energy and water balance** | | | |
| Survival from the onset of the study (%, n) | 66% (6/9) | 100% (9/9) | 0.07 # |
| Body mass (g) | 475.2 ± 17.1 | 476.3 ± 12.1 | 0.43 |
| 24 hr food intake (g) | 23.2 ± 3.2 | 24.2 ± 2.3 | 0.26 |
| 24 hr water intake (mL) | 37.5 ± 7.5 | 41.1 ± 6.6 | 0.17 |
| 24 hr urine output (mL) | 14.8 ± 2.8 | 17.9 ± 2.5 | **0.02** |
| Tibia length (cm) | 3.95 ± 0.21 | 3.99 ± 0.11 | 0.34 |
| **TMAO** | | | |
| Plasma TMAO (μmol/L) | 6.71 ± 1.49 | 20.32 ± 7.21 | **<0.001** |
| 24 hr TMAO urine excretion (μmoles) | 9.97 ± 3.46 | 126.8 ± 32.8 | **<0.001** |
| **Heart mass** | | | |
| Heart mass (g) | 1.87 ± 0.31 | 1.72 ± 0.3 | 0.19 |
| **Arterial blood pressure and heart rate** | | | |
| Systolic (mmHg) | 136.2 ± 12.8 | 126.8 ± 12.7 | 0.11 |
| Diastolic (mmHg) | 98.6 ± 7.3 | 87.6 ± 5.6 | **0.004** |
| HR (beats/min) | 314 ± 61 | 302 ± 20 | 0.31 |
| **Echocardiographic parameters** | | | |
| LVEDV (mL) | 0.37 ± 0.19 | 0.52 ± 0.20 | 0.11 |
| LVESV (mL) | 0.14 ± 0.08 | 0.15 ± 0.1 | 0.41 |
| IVSs (cm) | 0.41 ± 0.01 | 0.35 ± 0.09 | 0.21 |
| IVSd (cm) | 0.29 ± 0.05 | 0.27 ± 0.07 | 0.28 |
| SV (mL) | 0.24 ± 0.12 | 0.36 ± 0.12 | 0.053 |
| EF (%) | 64 ± 8.5 | 71 ± 6.1 | 0.06 |
| **Left ventricle hemodynamic parameters (direct measurements)** | | | |
| LVEDP (mmHg) | 3.10 ± 0.78 | 3.41 ± 1.95 | 0.87 |
| dP/dt (mmHg/ms) | 5.88 ± 0.92 | 5.50 ± 0.98 | 0.41 |
| -dP/dt (mmHg/ms) | 2.35 ± 0.28 | 2.55 ± 0.67 | 0.27 |
| **Plasma NT-proBNP** | | | |
| NT-proBNP (pg/mL) | 52.26 ± 0.15.0 | 42.80 ± 9.5 | 0.09 |
| Electrolyte balance | | | |
| Serum sodium (mmol/L) | 142.4 ± 3.31 | 138.9 ± 2.98 | **0.04** |
| 24 hr sodium urine excretion (mmoles) | 1.42 ± 0.28 | 1.93 ± 0.33 | **0.005** |
| Serum potassium (mmol/L) | 4.73 ± 0.33 | 4.49 ± 0.28 | 0.09 |
| 24 hr potassium urine excretion (mmoles) | 2.89 ± 0.42 | 3.40 ± 0.54 | **0.04** |
| Serum creatinine clearance (mL/min) | 0.42 ± 0.17 | 0.53 ± 0.05 | 0.06 |
| **Hormones** | | | |
| Angiotensin II (pg/mL) | 325.7 ± 39.8 | 276.7 ± 38.3 | **0.02** |
| Aldosterone (pg/mL) | 816.8 ± 300.4 | 758.4 ± 142.8 | 0.32 |
| Vasopressin (ng/mL) | 3.02 ± 1.24 | 3.11 ± 1.03 | 0.45 |

*Table 2 continued on next page*

*Table 2 continued*

| Group/Parameter | SHHF-control | SHHF-TMAO | P |
|---|---|---|---|
| Cytokines | | | |
| TNF-a (pg/mL) | 34.56 ± 24.69 | 24.98 ± 7.92 | 0.19 |
| IL-10 (pg/mL) | 15.91 ± 4.66 | 28.17 ± 14.39 | **0.036** |

The online version of this article includes the following source data for Table 2:

**Source data 1.** Metabolic, renal and cardiovascular parameters in SHHF rats.

Kit. The absorbance intensity was measured at 450 nm with a Multiskan Microplate Reader (Thermo Fisher Scientific). All experiments were performed in duplicate (technical replicates).

## Histopathological evaluation

Tissues sections were fixed in 10% buffered formalin, dehydrated using graded ethanol and xylene baths and embedded in paraffin wax. Sections of 3–4 µm were stained with hematoxylin and eosin (HE) and van Gieson stain (for connective tissue fibers). General histopathological examination was evaluated at a magnification of 10x, 40x and 100x (objective lens) and 10x (eyepiece) and photographic documentation was taken. Morphometric measurements were performed at magnification of 40x (objective lens).

## Molecular biological procedures

Heart and kidney samples were collected from rats under urethane anesthesia and frozen at −80°C. Next, the samples were homogenized on BeadBug microtube homogenizer (Benchmark Scientific, Inc). Total RNA was isolated from the samples according to the TRI Reagent protocol. cDNA was transcribed from RNA samples according to the iScript Reverse Transcription Supermix #1708841 protocol (Bio-Rad). The qPCR mixes were prepared according to the Bio-Rad SsoAdvanced Universal SYBR Green Supermix protocol #1725271. Amplifications were performed on a Bio-Rad CFX Connect Real-Time System under standardized conditions using commercial assays. Data were analyzed using CFX Manager 3.0 software. The genes investigated in this study were: angiotensinogen (*Atg*, qRnoCED0051666), angiotensin II receptor type 1a (*At1a*, qRnoCID0052626), angiotensin II receptor type 1b (*At1b*, qRnoCED0005729), angiotensin II receptor type 2 (*At2*, qRnoCED0007551), transforming growth factor-beta (*Tgf-b*, qRnoCED0007638), renin (*Rn*, qRnoCID0008721), metalloproteinase inhibitor 2 (*Timp2*, qRnoCID0001559). Beta-actin was used as housekeeping gene (*Actbl2*, qRnoCED0018219).

## The effect of TMAO on diuresis, acute experiments

### Surgical preparations

Male SD rats were anaesthetized with urethane 1.5 g/kg b.w. i.p., which provided stable anesthesia for at least 4 hr. The jugular vein was cannulated for fluid infusions, and the carotid artery for ABP measurement with the Biopac MP 150 (Biopac Systems, USA). The bladder was exposed with an abdominal incision and was cannulated for timed urine collection. After the surgery, 20–30 min was allowed for stabilization. During this time, 0.9% saline was infused intravenously at a rate of 5 ml/kg/h. After completion of all experiments, the rats were euthanized by decapitation and both kidneys were excised and weighed.

### Experimental procedures and measurements

At the end of the stabilization period, three or four 10 min urine collections were taken to determine baseline water, sodium and total solute excretion rates in each experimental group. After stabilization of urine flow, TMAO (n = 8) was infused first as a priming dose of 2.8 mmol/kg b.w. in 5 mL / kg b.w. of 0.9% saline/5 min, followed by an infusion delivering 2.8 mmol/kg b.w. of TMAO in 5 mL/kg b.w of saline/60 min. At the start of the priming injection, five 10 min urine collections were taken during the infusion of TMAO. This basic protocol was applied in the two following protocols where TMAO was replaced by its solvent (0.9% NaCl) or saline solution of urea (2.8 mmol/kg).

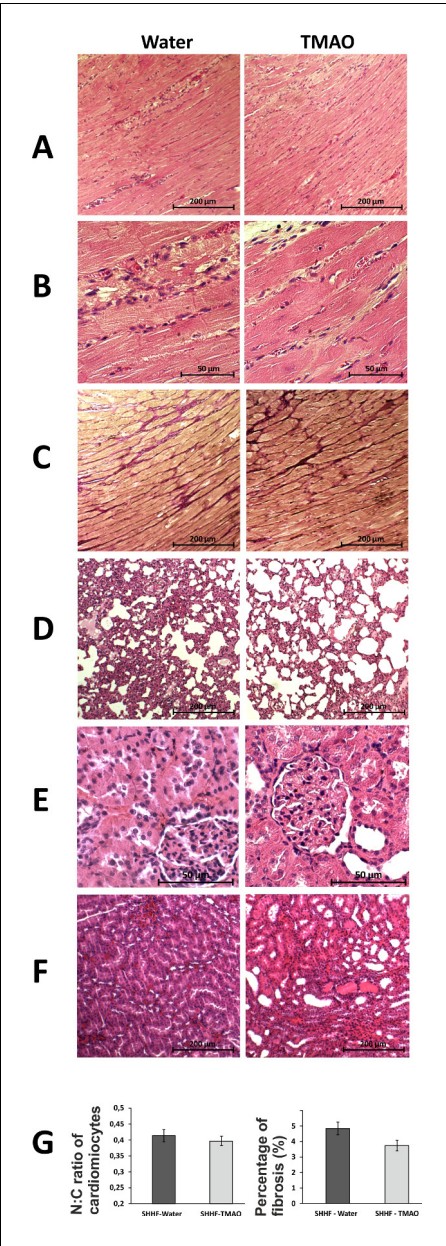

**Figure 6.** Histopathological picture of the heart, lungs and kidneys in Spontaneously Hypertensive Heart Failure (SHHF/MccGmiCrl-*Lepr^cp*/Crl) drinking either water (control group) or TMAO solution. A - myocardium; hematoxylin-eosin staining at magnification x10; B - myocardium; hematoxylin-eosin staining at magnification x40; C - myocardium; van Gieson staining at magnification x10; D - lungs; hematoxylin-eosin staining at magnification x10; E – kidney - renal cortex, renal bodies; hematoxylin-eosin staining at magnification x40; F – kidney - renal medulla; hematoxylin-eosin staining at magnification x10; G – Percentage of myocardial fibrosis [%], N:C ratio of cardiomyocytes, values are means,± SE.

The online version of this article includes the following source data for figure 6:

**Source data 1.** Percentage of myocardial fibrosis in SHHF rats.

i.   Effect of drug solvent infusion (n = 5). These experiments served as a control for the equivalent volume administration of fluid bolus.

ii.  Effect of hypertonic fluid infusion of urea (n = 6). These experiments served as a control for equivalent volume administration of equivalent hypertonic fluid. The urea solution was equimolar with the solution of TMAO.

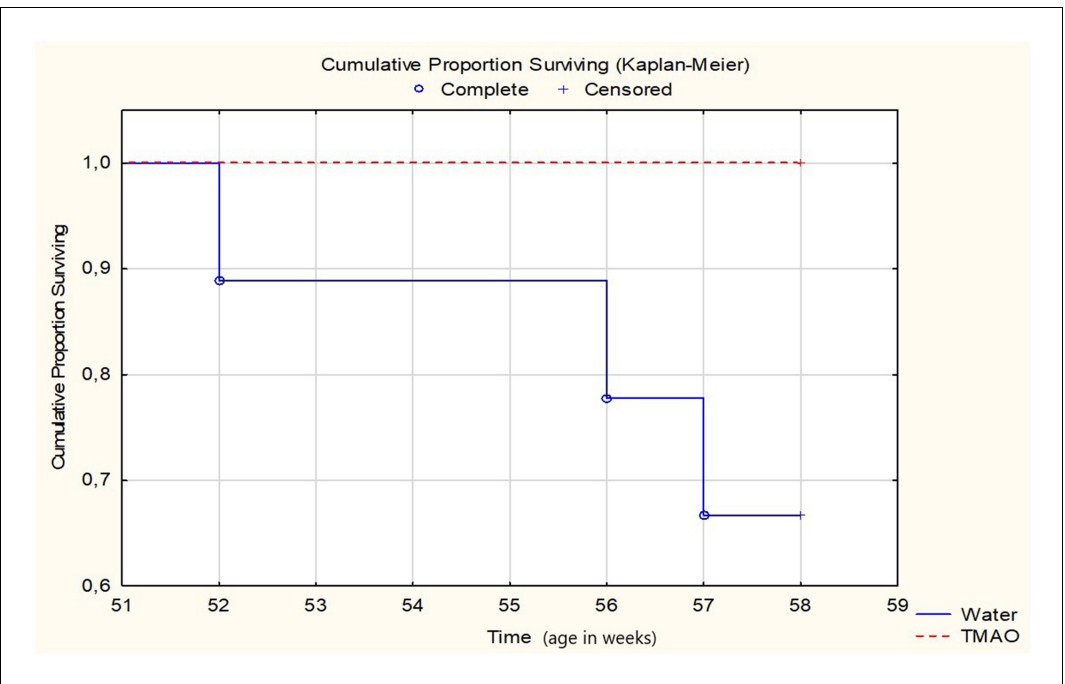

**Figure 7.** Survival Kaplan-Meier curves for SHHF - Spontaneously Hypertensive Heart Failure (SHHF/MccGmiCrl-*Lepr*$^{cp}$/Crl) rats drinking either water (control group, n = 9) or TMAO solution (n = 9). Log-Rank test p=0.06555.

## Analytical procedures and calculations

Urine volume was determined gravimetrically. Urinary osmolality (Uosm) was measured with the cryoscopic Osmomat 030 osmometer (Gonotec, Berlin, Germany). Urine sodium (UNa) and potassium (UK) concentration were measured by a flame photometer (BWB-XP, BWB Technologies Ltd, Newbury, UK). Urine flow (V), the excretion of total solutes (UosmV), sodium (UNaV) and potassium (UKV) were calculated using the standard formulas and standardized to g kidney weight (UXV/g KW). All measurements were performed in duplicates (technical replicates).

## Oscillatory-pressure controller and fluorescence correlation spectroscopy

We evaluated the effect of TMAO on bovine, cardiac LDH (Merck, Poland) exposed to pressure oscillations and increased temperature. The pressure oscillations were generated in a custom-built system. In order to mimic the conditions in the heart the pressure changed from 0 to 180–250 mmHg (or to higher values) at oscillation rate of 280 min-1. In general, the setup consisted of two main parts: i) a custom-built oscillatory pressure controller with solenoid micro valves to control the inner pressure and 'pulse' frequency and ii) a sample chamber (*Figure 11*). We designed and constructed three different samples chambers (*Figures 12*, *13* and *14*), which permitted the samples to be exposed to pressure oscillations in different ways.

### Oscillatory pressure controller

The custom-built oscillatory-pressure controller consisted of two pressure sources with constant but different air pressures (*p1* and *p2*, respectively). Each pressure source was connected through a plunger-type solenoid microvalve (V165, Sirai, Italy) to the inlet/outlet of the sample chamber via Teflon tubes (ID/OD = 0.8/1.6 mm, Bola, Germany) as shown in *Figure 11*. The two microvalves were controlled by a multiplexer switch module (National Instruments, USA) interfaced via custom-made LabView software which is freely available from the Github repository (*Nalin, 2020*). Due to their electro-magneto-mechanical construction the microvalves had some response time: the delay

**Table 3.** Metabolic, renal and cardiovascular parameters in 58-week-old SD-ISO rats.

Sprague-Dawley rats treated with isoprenaline at the age of 56 weeks. Rats maintained on either water (ISO-control, n = 5–9) or TMAO solution (ISO-TMAO, n = 7–10). T1 - metabolic and echocardiographic measurements, T2 - echocardiographic measurements, T3 - metabolic, echocardiographic and direct hemodynamic measurements (see also the study design, *Figure 1*). Creatinine clearance calculated as urine creatinine x urine output (ml/min)/plasma creatinine. LVEDV - left ventricle end diastolic volume, LVESV - left ventricle end systolic volume, SV – stroke volume, EF - ejection fraction, IVSs(d), intraventricular septum diameter during systole and diastole, respectively. LVEDP - pressure in the left ventricle during the end of diastole measured directly with a catheter, +dP/dt - maximal slope of systolic ventricular pressure increment, - dP/dt - maximal slope of diastolic ventricular pressure decrement. Values are means,± SD. P values by t-test or Mann-Whitney U test, except # - by log-rank test.

| Group/Parameter | | ISO-control | ISO-TMAO | P |
|---|---|---|---|---|
| **Survival, Energy and water balance** | | | | |
| Survival from the study onset (%, n) | | 90% (9/10) | 100% (10/10) | 0.32# |
| Body mass (g) | T1 | 434.44 ± 22.93 | 432.69 ± 37.59 | 0.45 |
| | T3 | 438.67 ± 23.99 | 428.71 ± 37.14 | 0.25 |
| 24 hr food intake (g) | T1 | 21.54 ± 1.43 | 21.84 ± 1.94 | 0.51 |
| | T3 | 21.24 ± 2.44 | 22.08 ± 1.91 | 0.21 |
| 24 hr water intake (mL) | T1 | 34.99 ± 3.10 | 34.82 ± 3.89 | 0.53 |
| | T3 | 34.77 ± 5.47 | 35.59 ± 1.92 | 0.34 |
| 24 hr urine output (g) | T1 | 19.28 ± 2.73 | 20.03 ± 4.42 | 0.34 |
| | T3 | 19.68 ± 4.29 | 20.23 ± 2.6 | 0.37 |
| Tibia length (cm) | T3 | 4.33 ± 0.06 | 4.34 ± 0.1 | 0.35 |
| **TMAO** | | | | |
| Plasma TMAO (µmol/L) | T3 | 5.95 ± 2.35 | 32.51 ± 11.43 | **<0.001** |
| 24 hr TMAO urine excretion (µmoles) | T3 | 5.74 ± 1.64 | 119.12 ± 65.96 | **<0.001** |
| **Heart mass** | | | | |
| Heart mass (g) | T3 | 1.46 ± 0.13 | 1.43 ± 0.19 | 0.39 |
| **Arterial blood pressure and heart rate** | | | | |
| Systolic (mmHg) | T3 | 142.48 ± 10.63 | 130.92 ± 11.67 | **0.026** |
| Diastolic (mmHg) | T3 | 97.10 ± 9.95 | 87.59 ± 10.47 | **0.037** |
| HR (beats/min) | T3 | 356.56 ± 23.39 | 344.19 ± 49.69 | 0.26 |
| **Echocardiographic parameters** | | | | |
| LVEDV (mL) | T1 | 0.44 ± 0.22 | 0.33 ± 0.17 | 0.15 |
| | T2 | 0.54 ± 0.22 | 0.53 ± 0.23 | 0.47 |
| | T3 | 0.51 ± 0.11 | 0.53 ± 0.34 | 0.21 |
| LVESV (mL) | T1 | 0.13 ± 0.09 | 0.11 ± 0.07 | 0.31 |
| | T2 | 0.22 ± 0.13 | 0.13 ± 0.05 | 0.054 |
| | T3 | 0.12 ± 0.05 | 0.08 ± 0.02 | **0.01** |
| IVSs (cm) | T1 | 0.33 ± 0.03 | 0.32 ± 0.05 | 0.29 |
| | T2 | 0.33 ± 0.06 | 0.35 ± 0.09 | 0.29 |
| | T3 | 0.36 ± 0.04 | 0.32 ± 0.05 | 0.07 |
| IVSd (cm) | T1 | 0.24 ± 0.04 | 0.23 ± 0.04 | 0.28 |
| | T2 | 0.26 ± 0.03 | 0.25 ± 0.04 | 0.41 |
| | T3 | 0.24 ± 0.04 | 0.25 ± 0.03 | 0.31 |
| SV (mL) | T1 | 0.29 ± 0.15 | 0.26 ± 0.12 | 0.40 |
| | T2 | 0.35 ± 0.17 | 0.47 ± 0.22 | 0.09 |
| | T3 | 0.38 ± 0.09 | 0.34 ± 0.07 | 0.69 |

*Table 3 continued on next page*

*Table 3 continued*

| Group/ Parameter | | ISO-control | ISO-TMAO | P |
|---|---|---|---|---|
| EF (%) | T1 | 71.22 ± 6.12 | 72.11 ± 14.51 | 0.45 |
| | T2 | 73.11 ± 8.88 | 70.11 ± 12.84 | 0.28 |
| | T3 | 78.67 ± 7.18 | 77.89 ± 11.94 | 0.43 |
| Left ventricle hemodynamic parameters (direct measurements) | | | | |
| LVEDP (mmHg) | T3 | 6.73 ± 2.55 | 4.46 ± 0.74 | **0.03** |
| dP/dt (mmHg/ms) | T3 | 9.20 ± 1.54 | 6.55 ± 1.18 | **0.004** |
| -dP/dt (mmHg/ms) | T3 | 5.19 ± 0.38 | 4.76 ± 0.65 | 0.09 |
| Plasma NT-proBNP | | | | |
| NT-proBNP (pg/mL) | T3 | 64.49 ± 43.59 | 22.01 ± 22.83 | **0.02** |
| Electrolyte balance | | | | |
| Serum sodium (mmol/L) | T3 | 136.88 ± 3.56 | 137.89 ± 1.62 | 0.23 |
| 24 hr sodium urine excretion (mmoles) | T3 | 1.93 ± 0.27 | 2.09 ± 0.42 | 0.18 |
| Serum potassium (mmol/L) | T3 | 5.53 ± 0.89 | 5.02 ± 0.77 | 0.11 |
| 24 hr potassium urine excretion (mmoles) | T3 | 2.56 ± 0.35 | 2.92 ± 0.35 | **0.03** |
| Serum creatinine clearance (mL/min) | T3 | 1.26 ± 0.23 | 1.24 ± 0.26 | 0.43 |
| Hormones | | | | |
| Angiotensin II (pg/mL) | T3 | 286.4 ± 24.4 | 272.6 ± 39.5 | 0.35 |
| Aldosterone (pg/mL) | T3 | 938.6 ± 114.6 | 1032.4 ± 120.6 | 0.07 |
| Vasopressin (ng/mL) | T3 | 0.98 ± 0.55 | 1.28 ± 0.66 | 0.18 |

The online version of this article includes the following source data for Table 3:

Source data 1. Metabolic, renal and cardiovascular parameters in SD-ISO rats.

between application of the current and opening of the valve was 24 ms, and the delay between stopping the current and shutting the valve was 8 ms (*Churski et al., 2010*). The time shifts were taken into consideration when calculating the required oscillatory pressure.

Operation protocol: The two valves were initially closed and the pressure inside the sample chamber was equal to the atmospheric pressure. Upon initiating the oscillatory system, the first valve opened for 50 ms. As the first valve closed, the second valve opened for 50 ms followed by a 50 ms pause when both valves were closed. With the response times included, one full cycle took 214 ms, generating an oscillation of ~280/minute between pressure p1 and p2, mimicking the heartbeat of a rat. This pressure-oscillatory system was used in each relevant experiment with the only difference being the pressure applied: p1 and p2.

## Microfluidic heart chip

In the first set of experiments a microfluidic device was constructed in polydimethylsiloxane (PMDS; Sylgard 184, Dow Corning, USA) in three steps; first, the channels and sample chamber (in the shape of a heart) were micro-milled in a polycarbonate plate (PC; Macrolon, Germay) using a CNC milling machine (MSG4025, Ergwind, Poland); second, we poured PDMS onto the PC chip, polymerized the PDMS at 75°C for 2 hr, activated the surface using a Laboratory Corona Treater (BD 20AC, Electro-Technic Products, USA) and silanized in the vapors of tridecafluoro-1,1,2,2-tetrahydrooctyl-1-trichlor-osilane (United Chemical Technologies, USA) for 60 min under vacuum (10mbar); third, this negative mold was used to produce the final PDMS chip. Inlet and outlet holes were punched using a small diameter (ID = 0.8 mm) biopsy puncher prior to bonding the PDMS chip to 1mm-thick glass slides using oxygen plasma for 45 s.

Operation parameters. The inlets were connected to the heart-shaped sample chamber (height of 300 μm and total volume of ~220 μL) via microchannels from two sides (*Figure 12a*). On one side, the microchannel ended in a single inlet through which the sample liquid was injected. Once the chamber was completely filled, the inlet was sealed air-tight. On the other side of the chamber, the

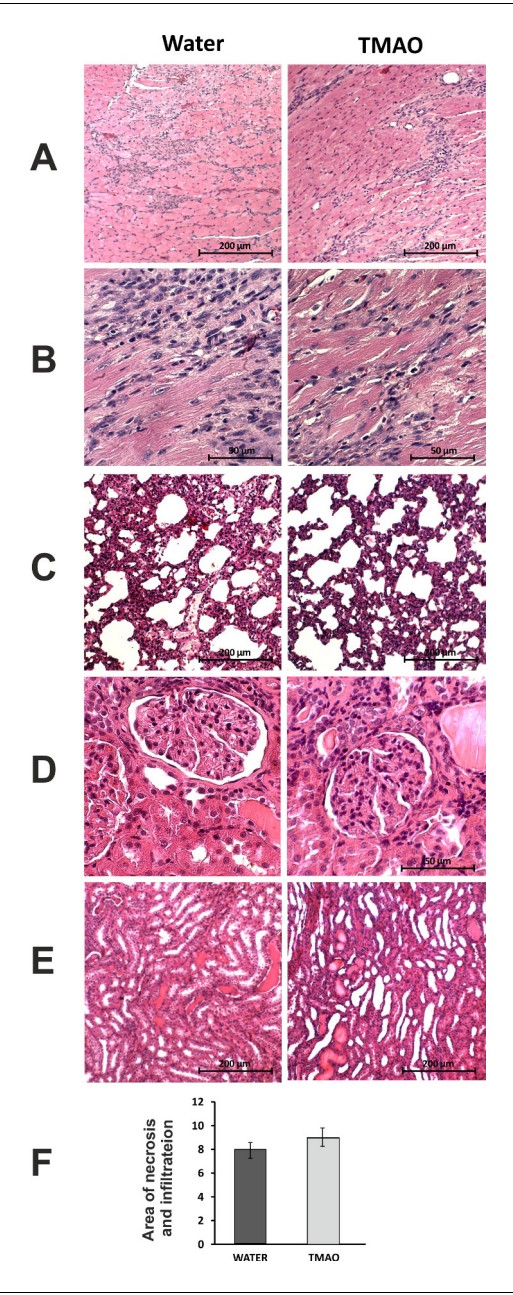

**Figure 8.** Histopathological picture of the heart, lungs and kidneys in Sprague-Dawley rats treated with isoprenaline at the age of 56 weeks, and drinking either water (control group) or TMAO solution. A - myocardium; hematoxylin-eosin (HE) staining at magnification x10; B - myocardium; HE staining at magnification x40; C - lungs; HE staining at magnification x10; D – kidney - renal cortex, renal bodies; HE staining at magnification x40; E – kidney - renal medulla; HE staining at magnification x10; F – Percentage of necrotic and inflammatory area in myocardium [%], values are means,± SE.

The online version of this article includes the following source data for figure 8:

**Source data 1.** Percentage of necrotic and inflammatory area in myocardium of SD-ISO rats.

microchannel branched into two channels, ending in 1–1 inlets. The two ducts were connected to two pressure sources (p1 and p2) of the oscillatory system by Teflon tubes (ID = 0.8 mm, OD = 1.6 mm; Bola, Germany). The valves were initially closed, and the atmospheric pressure acted on the sample.

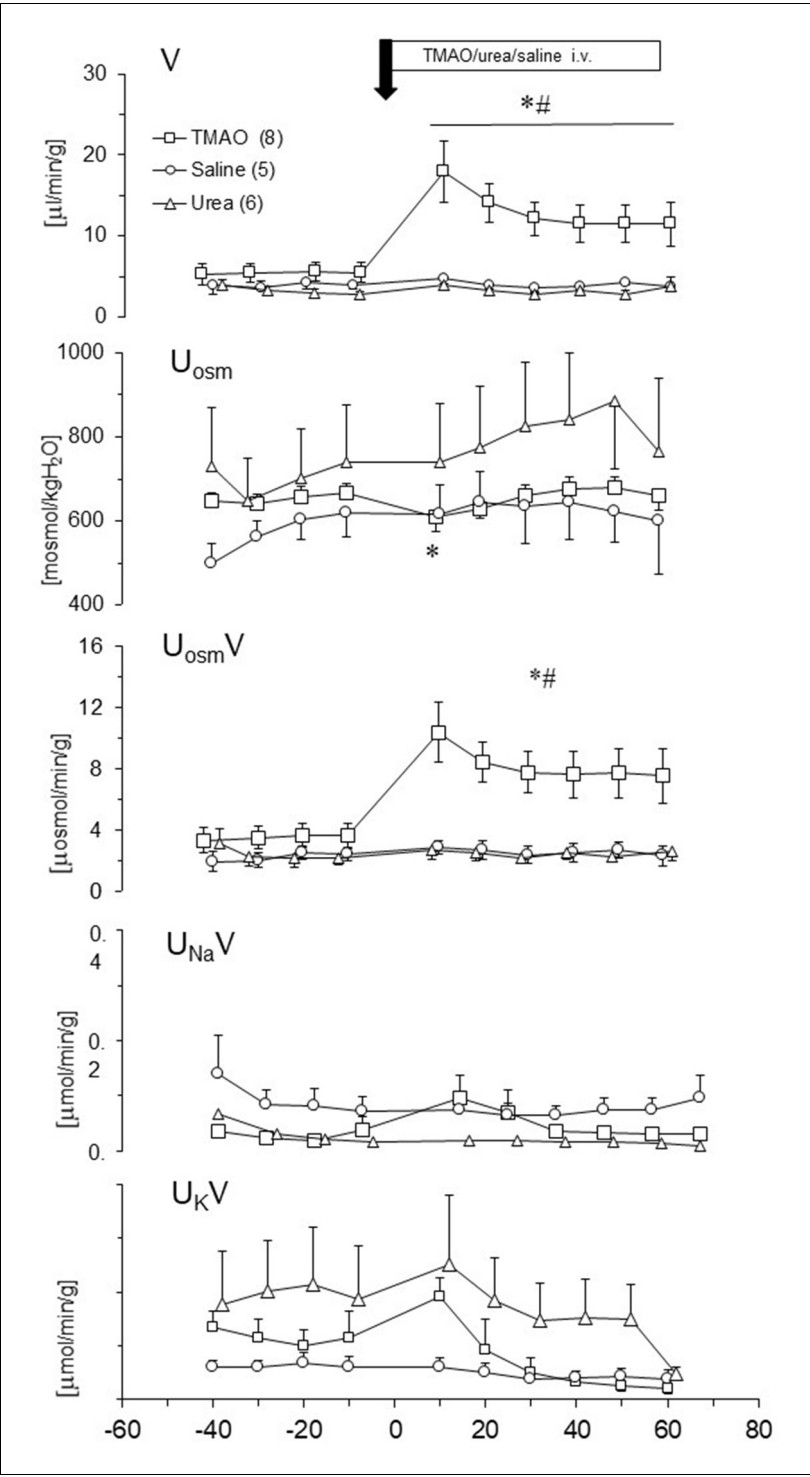

**Figure 9.** Effects of TMAO (n = 8), urea (n = 6) and saline (n = 5) on renal excretion in anesthetized Sprague-Dawley rats. The priming dose (indicated by arrow) of TMAO and urea were 2.8 mmol/kg b.w. (bolus), followed by continuous infusion at a rate of 2.8 mmol/kg b.w./60 min. V – urine flow; $U_{osm}$ – urine osmolality; $U_{osm}V$, $U_{Na}V$, $U_{K}V$ – total solute, sodium and potassium excretion, respectively. Values are means ± SE. * - $p<0.05$ vs pretreatment values, # - $p<0.02$ TMAO vs saline, TMAO vs urea.

The online version of this article includes the following source data for figure 9:

**Source data 1.** Effects of TMAO, urea and saline on renal excretion.

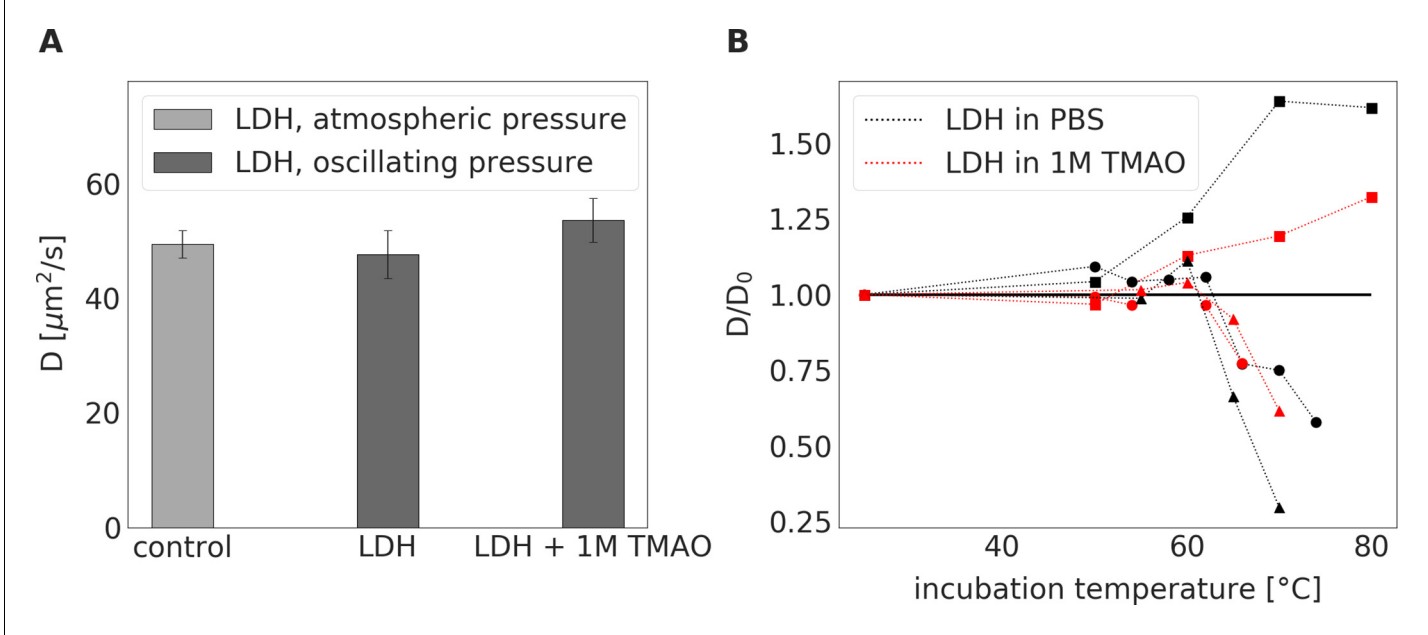

**Figure 10.** LDH incubated at a constant atmospheric pressure for 24 hrs or LDH exposed to oscillating pressure for 24 hrs with or without TMAO. (**A**) - Diffusion coefficient of LDH incubated either in PBS buffer at a constant atmospheric pressure and room temperature (serving as a control) or exposed to pressure oscillation for 24 hrs either in PBS buffer or in 1 M TMAO solution in PBS buffer at room temperature. Irrespective of the presence of 1M TMAO, 24-hour incubation under oscillating pressure did not cause dissociation, denaturation, or aggregation of LDH. (**B**) – Relative diffusion coefficient (diffusion coefficient $D$ divided by its value in PBS buffer at room temperature $D_0$) of LDH exposed to elevated temperatures for 15 min either in PBS buffer (black symbols) or 1 M TMAO solution in PBS (red symbols). Symbol shapes differentiate between three independent measurement series (series 1 (3nM of LDH) – squares, series 2 (30nM of LDH) – circles, series 3(300nM) – triangles). In series 1, we observed an increase in the relative diffusion coefficient, suggesting degradation of LHD tetramer structure to monomers. In series 2 and 3, an increase in diffusion coefficient was followed by a decrease in diffusion coefficient suggesting the degradation of LDH followed by the aggregation of the LDH monomers. The presence of TMAO shifted the threshold of change in relative diffusion coefficient towards higher temperatures.

The online version of this article includes the following source data and figure supplement(s) for figure 10:

**Source data 1.** LDH incubated at a constant atmospheric pressure for 24 hrs or LDH exposed to oscillating pressure for 24 hrs with or without TMAO.
**Figure supplement 1.** Diffusion coefficients of LDH measured by FCS.

In the first run, we applied p1 = 250 mmHg overpressure and p2 = 0 mmHg (i.e. atmospheric pressure) and oscillation frequency of ~280/minute as described above.

The inlet channel (from the side of the oscillatory system) was filled with the sample liquid only half-way. As the pressure was oscillating in the microfluidic device, rapid movement of the liquid meniscus could be observed according to the oscillation frequency (see *Figure 12b–d*). The over-pressure p1 pushed the liquid meniscus towards the chamber, while upon switch to the low pressure (p2) the meniscus pulled back. Unfortunately, since the liquid was in direct contact with the oscillating air some evaporation occurred. As a consequence, the sample liquid evaporated from the chamber within approximately 40 min of starting the experiment, leaving the microfluidic chamber dry.

## PDMS sample chamber in the shape of a micro centrifuge-tube

To avoid evaporation, we constructed a different microfluidic device, where the sample liquids were not in direct contact with the oscillating air. Briefly, we removed the caps of four 0.5 mL conical bottom micro centrifuge tubes (Eppendorf, Germany) and glued them to a microscope glass slide (75 × 24×1 mm) in a line separated by ~3 mm from each other. On two other, larger glass plates (75 × 50×1 mm), we glued small blocks of polycarbonate (45 × 15×5 mm) and fixed them on two opposing sides of the micro-centrifuge tube array, so that the distance between the polycarbonate blocks and the tubes from both sides was ~1 mm. We put this construction into a small box, filled it with polydimethylsiloxane (PDMS elastomer, Sylgard 184 mixed with curing agent at a ratio of 10:1 and degassed) and polymerized the PDMS at 75℃ for 2 hr. After that, we removed it from the container,

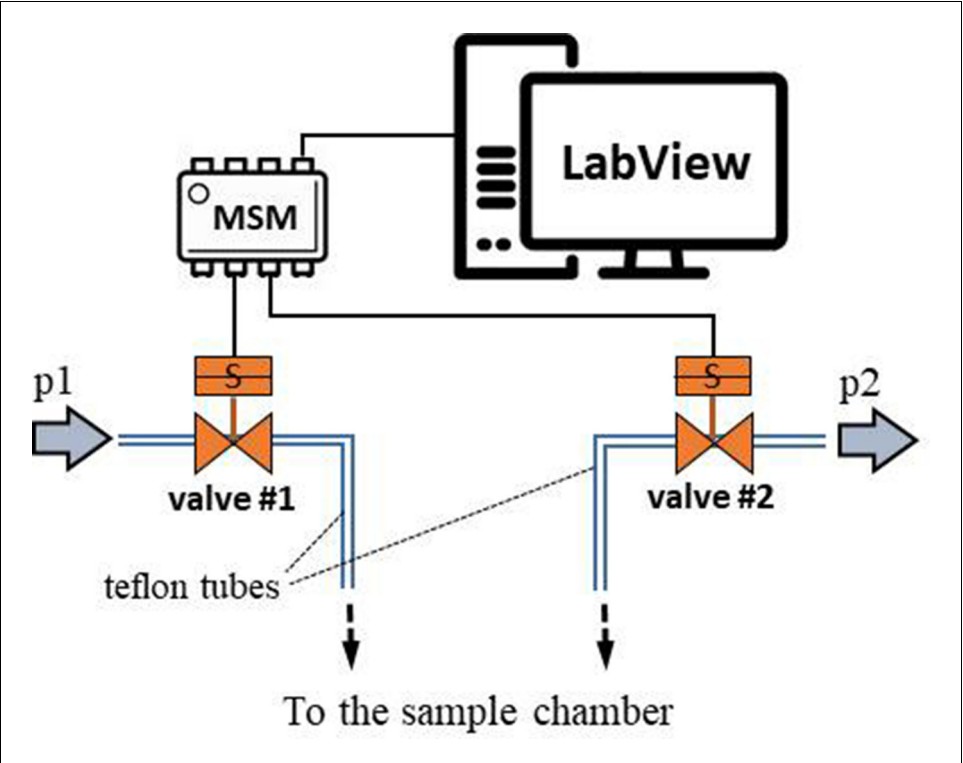

**Figure 11.** Schematic illustration of the oscillatory pressure controller.

removed the glass plates, the polycarbonate blocks and micro centrifuge tubes from the cured PDMS. Next, we bonded the PDMS block from the two sides, where the removed polycarbonate blocks left a cuboid-shaped cavity to glass plates (75 × 50×1 mm) using oxygen plasma. We also inserted steel capillaries (OD = 0.8 mm) from two sides of each cavity to which we connected the oscillatory system by Teflon tubes (Bola, Germany). Next, we constructed a 4 mm thick PDMS plate to cover the top of the chamber array, which was left open after the Eppendorf tubes were removed from the PDMS. Prior to bonding with oxygen plasma, holes were punched in the PDMS cover to allow for sample injection. Once the microdevice was ready (see *Figure 13d*), we injected the sample liquids through the punched inlet holes into the sample chambers and sealed the inlet holes airtight. The sample liquids filled the chamber completely. In a separate experiment we left small air bubbles in the Eppendorf-chambers for pressure estimation (see below).

Operation parameters. After the chambers were filled the oscillatory system was turned on using pressures p1 = 2.5 bar and p2 = 500 mbar (equivalent to 1875 and 375 mmHg, respectively). Pressure higher than 2.5 bar resulted in destruction of the PDMS membrane between the sample chambers and the pressurized cavity. As the high-pressure air filled the cuboidal-cavities, they expanded and pushed the 1 mm thick PDMS membrane towards the sample-filled chambers squeezing and deforming them (the glass on the other side of the cavity prevented the expansion) (see *Figure 13b–c*). As the valve of the high-pressure source (p1) closed and the second valve opened, the pressure in the cavity dropped significantly and the deformed sample chambers returned to their original shape. The rapid oscillation of the air pressure (~280/min) in the cavities resulted in a corresponding squeeze-release pulsation of the chambers, mimicking a beating heart better than previous method. After 24 hr of operation, the system was turned off, the plugs were removed from the inlet holes and the samples were removed with a syringe and needle for analysis.

## Pressure-bottle-pressure significantly exceeding the physiological range

We put four samples into four 0.5 mL micro-centrifuge tubes (Eppendorf tubes), removed the caps of the tubes and covered them with parafilm (Bemis, USA) instead. Small holes were made in the parafilm with a syringe needle (OD = 0.6 mm). We put the sample-loaded Eppendorf tubes inside a

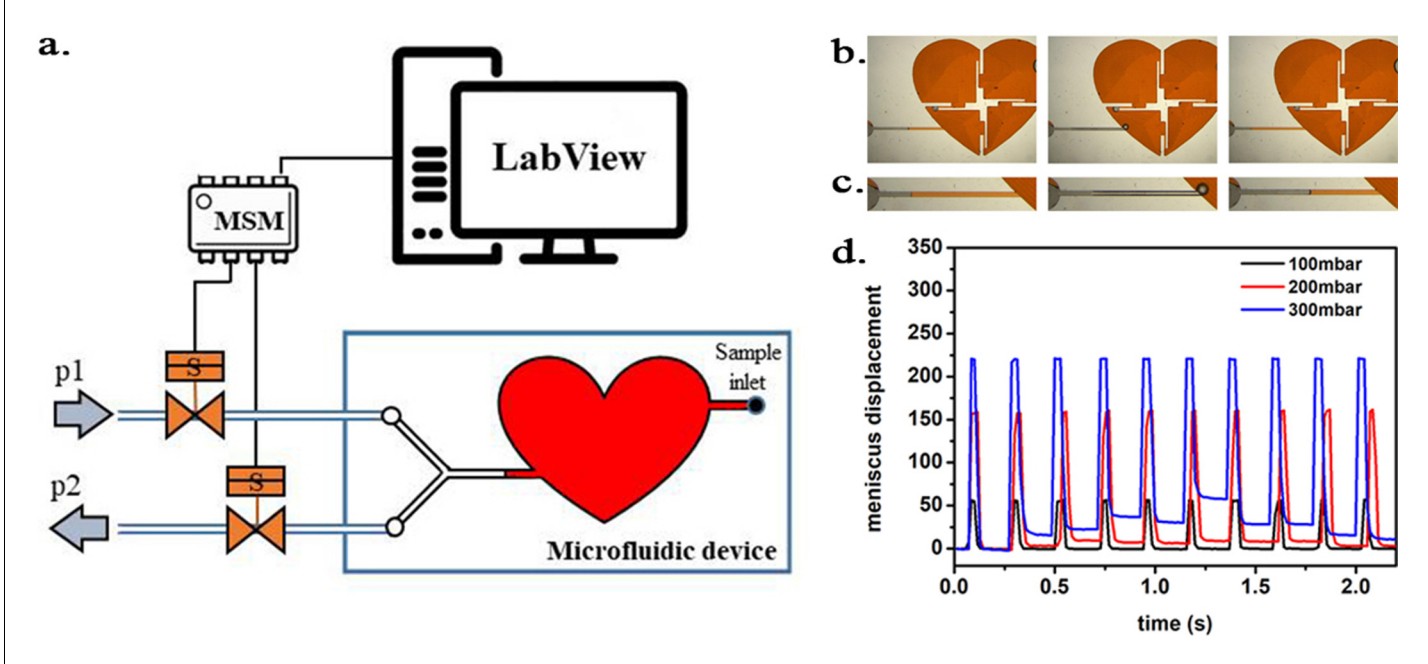

**Figure 12.** Scheme of the experimental setup. (**a**) Scheme of the experimental setup with the heart-shaped polydimethylsiloxane (PDMS) microfluidic device. (**b**) time-sequence snapshots of the device in operations. From left to right: the device is filled with liquid and is at rest (for better visualisation we used here red-dyed water instead of the transparent protein solution). The microchannel connecting the pressure system to the microfluidic chamber is filled halfway. Applying high pressure (valve # one open) pushes the liquid meniscus towards the chamber. Closing valve # one and opening valve # 2 (low pressure) the liquid meniscus pulls back (even further than its original position). (**c**) Close-up of the moving liquid meniscus, (**d**) oscillation profile generated from the position of the liquid meniscus in the microchannel as a function of time for various pressure differences Δp=p1 p2. The online version of this article includes the following source data for figure 12:

**Source data 1.** Oscillation profile generated from the position of the liquid meniscus in the microchannel.

250 mL laboratory glass bottle (Duran, Fischer Scientific, Germany) and closed it air-tight with a screw cap. Before closing, two small holes were drilled in the cap and steel capillaries (OD = 0.8 mm) were placed in and glued with an epoxy-based resin. The oscillatory system was connected to the steel capillaries with Teflon tubes.

Operation parameters. In this experiment, we decided to work well above the target pressure range, in order to increase the chance of protein denaturation. After a few pilot experiments, we set the high pressure p1 equal to 4.5 bar (3375 mmHg) and the lower pressure p2 equal to 0.5 bar (375 mmHg). However, due to the relatively large volume of the bottle, high compressibility of the air and the short opening times of the two valves, the pressures acting on the samples were not equal to the input pressures p1 and p2.

To measure the actual pressure range inside the laboratory bottle, which acted directly on the samples, we attached two manometers (model 82100, AZ Instruments, Taiwan) to the pressure out-let of the setup: one before the second valve to measure the pressure that build-up inside the laboratory bottle and another behind the second valve to measure the pressure after the release (*Figure 14*).

In the laboratory bottle the pressure acting on our samples never dropped below 450 mbar (320 mmHg) and increased up to 1600 mbar (1200 mmHg). The frequency oscillation of the pressure was ~230/min with some 'desynchronization' events every 15–20 s when the pressure difference Δp=p1 p2 was only ~150 mbar (110 mmHg). Whereas during synchronized operation, the pressure difference was ~600 mbar (450 mmHg).

## Proteins under hydrostatic pressure stress (HPS)

In all experiments, proteins at a concentration of around 1 µmol/L were incubated in the pressure oscillation system for 24 hr at room temperature. In each experimental run, we incubated in parallel

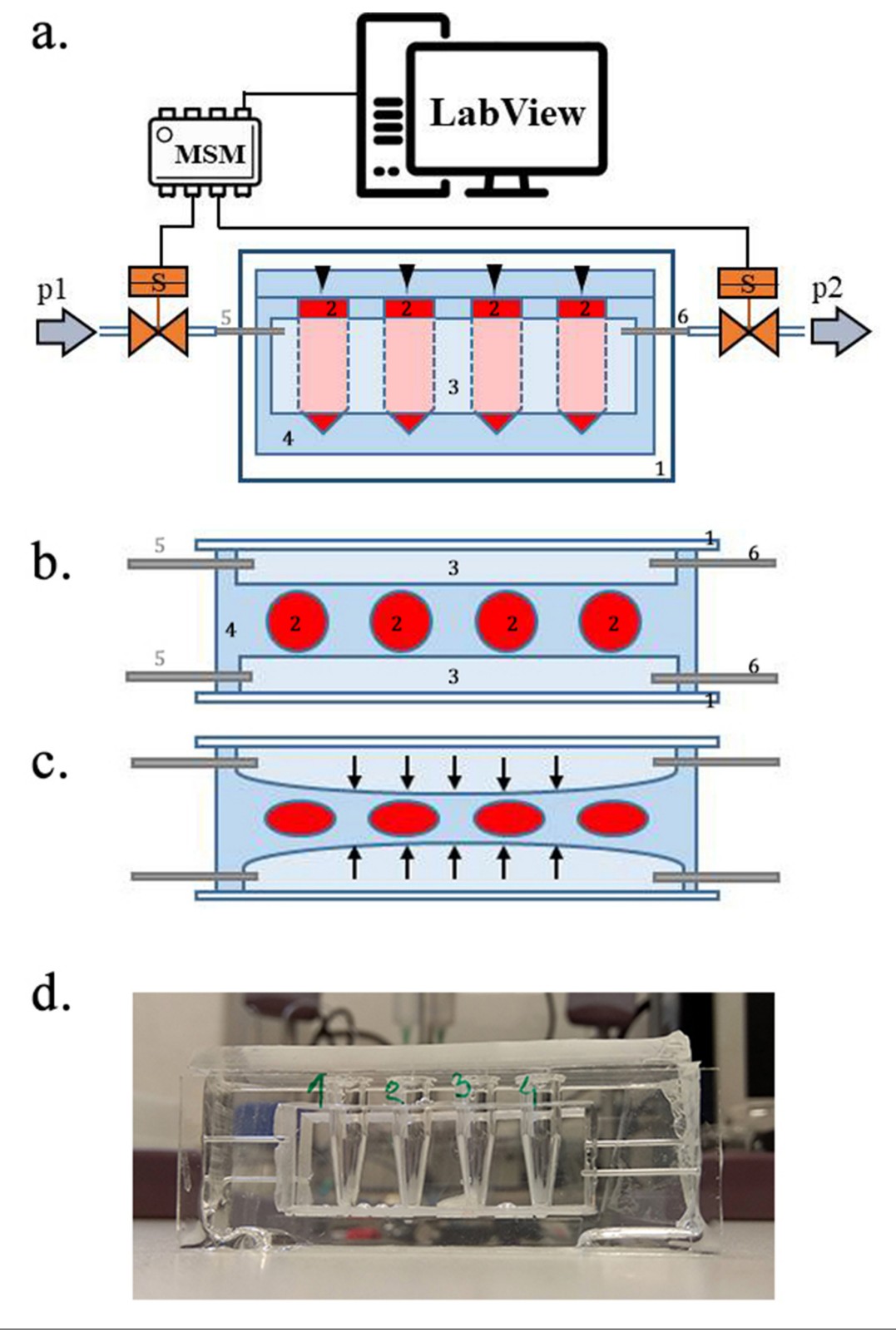

**Figure 13.** Scheme of the experimental setup. (a) Scheme of the set-up with the pressure oscillator connected to the Eppendorf chip with side view and b) top view of the 'Eppendorf chip'.Numbers indicate the glass slide (1), the sample chambers (2), the cuboidal cavity (3), the PDMS and the inlet (5) and outlet (6)*Stubbs et al., 2019* steel capillaries, respectively. (b–c) Schematic representation of the chip's operation upon applied pressure: the
*Figure 13 continued on next page*

*Figure 13 continued*

cavity (3) expands towards the sample chambers (2) squeezing and deforming them. (**d**) Photo of the constructed device prior to filling and connecting to the pressure controller.

proteins suspended in pure phosphate buffer saline (PBS) and 1M solution of TMAO in PBS. As a control for each of the samples, the solutions were divided into portions, where one portion was placed in a chamber exposed to oscillatory pressure while the other portion was placed in the same chamber but exposed to a constant, atmospheric pressure. All other conditions were the same for control and experimental samples. All solutions were then diluted 100-fold in PBS and fluorescence correlation spectroscopy (FCS) measurements were performed.

## Proteins under thermal stress

LDH solutions (3, 30, 300 nM), in either pure PBS or PBS supplemented with 1M TMAO, were pipetted into Eppendorf tubes. Samples were placed in a water heat bath (Lauda, electronically controlled) and incubated for 15 min (in separate tests – data not shown here – we verified that prolonging the incubation time above this limit, up to 1 hr, did not influence the results). Next, samples were immersed in room-temperature water bath to cool down and FCS measurement was performed for samples equilibrated to 25°C.

## Fluorescence correlation spectroscopy

The effect of hydrodynamic and thermal stress on Atto 488 (ATTO-TEC GmbH) labeled LDH structure was evaluated by means of FCS. Proteins were labeled using active NHS esters, according to a protocol supplied by the manufacturer. A 10-fold excess of the dye was used to ensure a high degree of labeling (which was especially important for LDH, where we intended to have at least one label per subunit to be able to monitor probe concentration changes upon dissociation of the

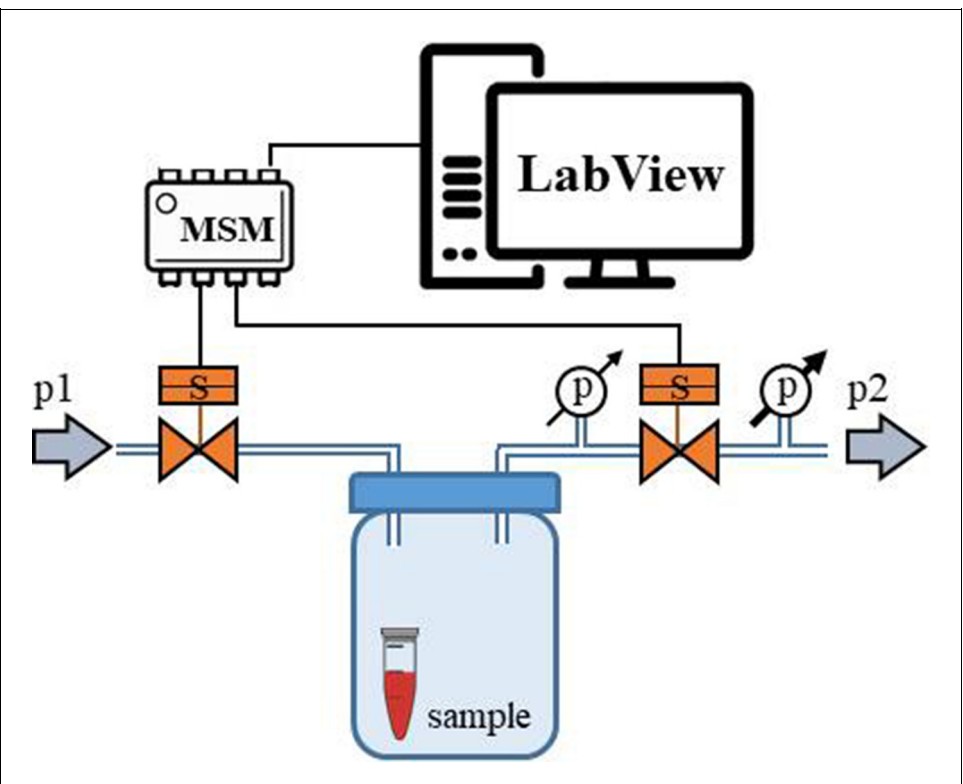

**Figure 14.** Schematic representation of the 'pressure bottle' system. Two manometers were attached to the system and used to determine the pressure acting on the samples.

tetramer). Post-labeling purification was performed using Bio-Rad BioGel P-30 size exclusion columns. Protein labeling and purification was performed immediately before starting each experimental run. Protein diffusion coefficients ($D$) are associated with the protein's hydrodynamic radius ($R_h$) by Stokes-Einstein relation, $D = k_B T / 6\pi\eta R_h$ (where $k_B$ is Boltzmann constant, $T$ is the temperature and $\eta$ denotes the viscosity of medium). The alteration in protein structure by dissociation of the quaternary structure results in an increase of the observed diffusion coefficient, while denaturation of tertiary structure causes a decrease of $D$ by a factor of $1.5 - 3$ (*Wilkins et al., 1999*).

FCS measurements were performed on dedicated FCS system, based on a Nikon C1 inverted confocal microscope (Nikon Instruments, Japan) with a PlanApo 60x, NA=1.2 water immersion objective. The setup is equipped with a Pico Harp 300 system (PicoQuant, Germany). Measurements were performed in a climate chamber (Okolab, Italy) providing temperature control and required humidity at $25.0 \pm 0.5 N oentity$. Labeled proteins were excited by a 485 nm laser, and fluorescence was detected through a 488/LP long-pass filter (Chroma, USA). Data acquisition and analysis was performed using SymPhoTime 64 software (PicoQuant, Germany). The experiments were preceded by establishing the dimension of the confocal volume using rhodamine 110 (Sigma-Aldrich, USA).

### Data analysis and statistics

Blinding was provided for each treatment (TMAO vs control). Unblinding was performed after statistical analysis. Any encountered outliers were included in the analysis. Diastolic arterial blood pressure (DBP), systolic arterial blood pressure (SBP) and heart rate (HR) were calculated from the arterial blood pressure tracing. Left ventricular end-diastolic pressure (LVEDP), maximal slope of systolic pressure increment (+dP/dt) and diastolic pressure decrement (-dP/dt) were calculated from the left ventricle blood pressure tracing using AcqKnowledge Biopac software (Biopac Systems, Goleta, USA). The Shapiro-Wilk test was used to test normality of the distribution. Differences between the TMAO and control groups were evaluated by an Independent-Samples t-test or by Mann-Whitney U test for data that were not normally distributed. In the acute experiments, the differences in the mean values between groups were first analyzed by the classic one-way ANOVA followed by a modified Student's t-test for independent variables, using Bonferroni's correction for multiple comparisons. The log-rank test was used to test the survival differences between TMAO and control animals. A value of two-sided p<0.05 was considered significant. Analyses were conducted using Statistica, version 13.3 (Tibco, Palo Alto, CA, USA).

### Replicates

Replicates were not used unless otherwise stated. The basic definitions of technical and biological replicates are as follows. Technical replicates: a test performed on the same sample multiple times. Biological replicates: a test performed on biologically distinct samples representing an identical time point or treatment dose.

### Sample size

Sample size was calculated at the start of the study, based on plasma levels of the investigated markers and hemodynamic parameters in rats, which were reported in our previous studies (*Savi et al., 2018*). We have chosen between-group difference in plasma NT-proBNP, ejection fraction, stroke volume and ABP as primary end-points with the following parameters, respectively: the difference between the tested (groups) 30%, 15%, 30% and 13%; the average for the entire population of 30 pg/mL, 80%, 0.35 mL, 100 mmHg; a common standard deviation of 7 pg/mL, 9%, 0.08 mL, 10 mmHg; for an alpha error of 0.05, test power 0.8. Other biochemical parameters were used as secondary end points.

## Acknowledgements

The authors are grateful to Tomasz Hutsch MDV, PhD, a veterinary pathologist, for consultations on histopathological analysis. LD thanks to Francesco Nalin for preparing the valve control program.

## Additional information

### Funding

| Funder | Grant reference number | Author |
|---|---|---|
| Narodowe Centrum Nauki | 2018/31/B/NZ5/00038 | Marcin Ufnal |

The funders had no role in study design, data collection and interpretation, or the decision to submit the work for publication.

### Author contributions

Marta Gawrys-Kopczynska, Marek Konop, Klaudia Maksymiuk, Katarzyna Kraszewska, Marta Pilz, Leszek Dobrowolski, Data curation, Formal analysis, Writing - original draft; Ladislav Derzsi, Krzysztof Sozanski, Robert Holyst, Conceptualization, Data curation, Formal analysis, Writing - original draft; Emilia Samborowska, Izabella Mogilnicka, Data curation, Formal analysis; Kinga Jaworska, Data curation, Formal analysis, Writing - original draft, Writing - review and editing; Marcin Ufnal, Conceptualization, Resources, Data curation, Formal analysis, Supervision, Funding acquisition, Investigation, Methodology, Writing - original draft, Project administration, Writing - review and editing

### Author ORCIDs

Marcin Ufnal (iD) https://orcid.org/0000-0003-0088-8284

### Ethics

Animal experimentation: The study was performed according to Directive 2010/63 EU on the protection of animals used for scientific purposes and approved by the Local Bioethical Committee in Warsaw (permission:100/2016 and 098/2019).

### Decision letter and Author response

Decision letter https://doi.org/10.7554/eLife.57028.sa1
Author response https://doi.org/10.7554/eLife.57028.sa2

## Additional files

### Supplementary files

- Transparent reporting form

### Data availability

All data generated or analysed during this study are included in the manuscript and supporting files. Source data files have been provided for all figures and tables.

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
