## [Decision Letter]

**Acceptance summary:**

The results of your study provide important mechanistic insights into the potential cardiovascular protective effects of TMAO in the context of heart failure.

**Decision letter after peer review:**

Thank you for submitting your article "TMAO, a seafood-derived molecule, produces diuresis and reduces mortality in heart failure rats" for consideration by *eLife*. Your article has been reviewed by three peer reviewers, including Arduino A Mangoni as the Reviewing Editor and Reviewer #1, and the evaluation has been overseen by a Senior Editor.

The reviewers have discussed the reviews with one another and the Reviewing Editor has drafted this decision to help you prepare a revised submission.

Summary:

This is a comprehensive study of TMAO effects on cardio-related effects in heart-failure rats. Much the work is interesting and useful, especially since TMAO from meat carnitine/microbiome was initially claimed to cause cardiovascular problems despite high TMAO in seafood, which may be cardioprotective. This study demonstrates possible protective effects.

1) Data:

– The authors found a beneficial effect of TMAO on SHHF, which is associated with diuretic, natriuretic and hypotensive effects. The authors explored the possible mechanism underlying the beneficial effects and found that it may be related to decreased RAS activity in heart by TMAO. However, this may not cover all the underlying mechanism because it has been shown that the SHHF is related to many mechanisms, such as increased sympathetic activity, inflammation in the periphery and brain except activation of the RAS. Is there evidence that TMAO can regulate the sympathetic activity, inflammation or other factors to reduce mortality of the SHHF rats by the authors or other researcher? The authors may add more discussion to clarify it.

– The authors found that TMAO was neutral for cardiac LDH protein structure. Is there change in LDH protein function after HS or TMAO? In other words, does the unaltered protein structure stand for the normal function of this protein in this study? Are there methods to measure the LDH protein function?

– LDH effects: a) Subsection “Effect of TMAO on structure of LDH exposed to HPS and increased temperature”/Figure 14B: Figure 14B is very confusing. The legend says it is 3 different series / runs, yet some graph lines go up (diffusion faster) while others go down (diffusion slower). Why? This needs clarification or interpretation in the Discussion. b) "In contrast, the protein stabilizing effect of TMAO is rather not involved in the protection of cardiac proteins against HPS produced by the contracting heart." While the data seem to support this conclusion for LDH in a limited way, there are hundreds of possible proteins that could be protected by TMAO in cycling pressures. Thus, this statement is far too sweeping and needs to be constrained.

– Pressure (HPS): Introduction paragraph four and elsewhere: The pressures used here such as 220 mmHg are nowhere near the levels in the deep sea where pressure significantly affects biological functions of deep-sea organisms. Moreover, pressure in the ocean is hydrostatic (from weight of unmoving column of water) while blood pressure is technically hydrokinetic (moving and pulsing) even though hydrostatic is commonly used. But this may matter since static and kinetic pressures could have very different effects on protein structure and adaptations. For example, in chondrocytes exposed to continuous high pressure, stress protein Hsp70 increased. However, cyclic pressure stress did not cause that (Kaarniranta et al., 1998). Thus, the analysis of TMAO on LDH under cycling pressures is not easily related to deep-sea TMAO/pressure studies for 2 reasons, namely i) mammalian pressures are far lower than deep-sea ones; and ii) mammalian pressures are kinetic, not static. I am not sure how to alter the paper for this, but some statement should be made.

– Diuresis: a) The primary benefits of TMAO here seem to be due to diuresis/natriuresis. What might the mechanism of this be, compared to other known diuretic agents? Discussion paragraph five suggests it is an *osmotic* effect, but this leads to:

b) Is this benefit specific to TMAO, or might *any* diuretic agent reduce blood pressure and thus have cardio benefits? This is not clear in this study – although urea did not cause diuresis (Figure 13), what about other diuretics? And since urea is known to cause diuresis, why did it not work here? – It is important to know if TMAO is very specific in cardioprotection or if many other agents would work as well.

c) Results subsection “SD rats”: TMAO rats had higher vasopressin, an anti-diuretic agent. Yet they also had higher diuresis. This seems to be apparent paradox – it should be interpreted in the Discussion, but I cannot find it.

2) Statistical analyses and date presentation:

– According to the authors only the (unpaired) t-test was used for statistical comparisons. Data have not been assessed for distribution normality (which would be required) and error bars in Figures 8, 9, 13 suggest non-normal distribution. All data need to be analyzed for normality of distribution, and accordingly parametric or non-parametric tests must be applied.

– Please indicate n number for each group in each figure panel or the corresponding figure legends.

– All P values in Tables 1, 2, and 3 need to be revised and amended, for example, a p<0.23 is meaningless.

– Figure 11: X-axis says "Time" but give only single numbers. It is assumed that this is to represent weeks. Please modify.

3) Language:

There are concerns regarding language. Some statements make no sense and need rewording, for example "TMAO reduces mortality in SHHF rats that is associated with diuretic, natriuretic and hypotensive effects. HPS and TMAO are neutral for cardiac LDH.", "Histopathological picture of SHHF heart corresponds", "in SHHF, lungs showed signs of passive hyperemia, with thickening of the interalveolar septa and a weak focal parenchymal edema.", "were euthanized or died spontaneously at the age of 52, 56 and 57-weeks due to hemiparalysis (ischemic stroke), dyspnoe (post-mortem lung edema) and spontaneous death (post-mortem lung edema), respectively." "MAO treated group showed… " "TMAO-treated rats showed a significantly lower diastolic blood pressure and a trend towards lower systolic blood pressure (p=0.01)”, "The general myocardium pattern"… etc. The authors are encouraged to consult with a scientist colleague whose first language is English and, where necessary, improve content, grammar, and syntax.

[Editors' note: further revisions were suggested prior to acceptance, as described below.]

Thank you for submitting your article "TMAO, a seafood-derived molecule, produces diuresis and reduces mortality in heart failure rats" for consideration by *eLife*. Your revised article has been reviewed by two peer reviewers, and the evaluation has been overseen by a Reviewing Editor and Matthias Barton as the Senior Editor.

The reviewers have discussed the reviews with one another and the Reviewing Editor has drafted this decision to help you prepare a further revised submission.

We would like to draw your attention to changes in our revision policy that we have made in response to COVID-19 (https://elifesciences.org/articles/57162). Specifically, we are asking editors to accept without delay manuscripts, like yours, that they judge can stand as *eLife* papers without additional data, even if they feel that they would make the manuscript stronger. Thus, the revisions requested below only address clarity and presentation.

Revisions:

The main concern is whether the manuscript has been reviewed by a native English speaker, as recommended. Although the language has been improved to some degree, there are still issues ("Water treatment", "Water group") which need correction. For example, the manuscript should always and only refer to a control group (currently "Water") or control (treatment).

The authors are requested to recheck all data normality using the Shapiro-Wilk test and revise the statistical analyses as required.

Was the water intake (the TMAO was added into the drinking water) quantified? How was this monitored and how could the authors ascertain the actual plasma TMAO concentration? Also, please indicate how TMAO affected water intake, which will be important for heart failure progression as well.

---

## [Author Response]

Revisions for this paper:1) Data:– The authors found a beneficial effect of TMAO on SHHF, which is associated with diuretic, natriuretic and hypotensive effects. The authors explored the possible mechanism underlying the beneficial effects and found that it may be related to decreased RAS activity in heart by TMAO. However, this may not cover all the underlying mechanism because it has been shown that the SHHF is related to many mechanisms, such as increased sympathetic activity, inflammation in the periphery and brain except activation of the RAS. Is there evidence that TMAO can regulate the sympathetic activity, inflammation or other factors to reduce mortality of the SHHF rats by the authors or other researcher? The authors may add more discussion to clarify it.

Thank you very much for this comment. Due to the multiple biochemical analyses which we have performed so far, we do not have sufficient samples left to fully address this issue. Nevertheless, we have managed to perform additional ELISA tests for IL-10, an anti-inflammatory cytokine, and TNF-α, a pro-inflammatory cytokine in plasma of SHHF-Water and SHHF-TMAO. In the TMAO group we found significantly higher plasma levels of IL-10 than in the Water group, there was also a trend towards higher level of TNF-α in the Water group (p=0.19). This has been added to the Table 2. To fully address the issue of sympathetic activity, further studies evaluating urine concentration of catecholamines metabolites are needed, but we are not able to perform these tests currently. According to the reviewer suggestions we have expanded the Discussion on this topic.

– The authors found that TMAO was neutral for cardiac LDH protein structure. Is there change in LDH protein function after HPS or TMAO? In other words, does the unaltered protein structure stand for the normal function of this protein in this study? Are there methods to measure the LDH protein function?

In our study, we have evaluated the effect of hydrostatic pressure stress (HPS) and TMAO on the structure but not on the activity of LDH. In general, changes in protein (enzyme) structure lead to disturbed (loss of) activity of proteins. However, our present findings only show changes to structure of LDH. Further studies are needed to evaluate LDH activity. We have highlighted this issue in the corrected manuscript.

– LDH effects: a) Subsection “Effect of TMAO on structure of LDH exposed to HPS and increased temperature”/Figure 14B: Figure 14B is very confusing. The legend says it is 3 different series / runs, yet some graph lines go up (diffusion faster) while others go down (diffusion slower). Why? This needs clarification or interpretation in the Discussion.

We have expanded the legend for the Figure 14B and discussion to clarify the interpretation of the results.

b) "In contrast, the protein stabilizing effect of TMAO is rather not involved in the protection of cardiac proteins against HPS produced by the contracting heart." While the data seem to support this conclusion for LDH in a limited way, there are hundreds of possible proteins that could be protected by TMAO in cycling pressures. Thus, this statement is far too sweeping and needs to be constrained.

This has been corrected.

“These in vitro experiments showed that changes in hydrostatic pressure generated by the contracting heart, and even much higher pressures, do not disturb the protein structure of cardiac LDH. (…) Nevertheless, the stabilizing effect of TMAO on other, less stable, cardiac proteins exposed to HPS cannot be excluded.”

– Pressure (HPS): Introduction paragraph four and elsewhere: The pressures used here such as 220 mmHg are nowhere near the levels in the deep sea where pressure significantly affects biological functions of deep-sea organisms. Moreover, pressure in the ocean is hydrostatic (from weight of unmoving column of water) while blood pressure is technically hydrokinetic (moving and pulsing) even though hydrostatic is commonly used. But this may matter since static and kinetic pressures could have very different effects on protein structure and adaptations. For example, in chondrocytes exposed to continuous high pressure, stress protein Hsp70 increased. However, cyclic pressure stress did not cause that (Kaarniranta et al., 1998). Thus, the analysis of TMAO on LDH under cycling pressures is not easily related to deep-sea TMAO/pressure studies for 2 reasons, namely i) mammalian pressures are far lower than deep-sea ones; and ii) mammalian pressures are kinetic, not static. I am not sure how to alter the paper for this, but some statement should be made.

We have added Discussion on this topic:

“The beneficial effects of TMAO appears to stem from its diuretic action rather than from the protein-stabilizing effect of TMAO on LDH, which was described for deep-sea animals, but was found not be involved here. […] In this regard, there is evidence that static pressure may have very different effects on cells and proteins than kinetic pressure (Kaarniranta et al., 1998).”

– Diuresis: a) The primary benefits of TMAO here seem to be due to diuresis/natriuresis. What might the mechanism of this be, compared to other known diuretic agents? Discussion paragraph five suggests it is an osmotic effect, but this leads to:b) Is this benefit specific to TMAO, or might any diuretic agent reduce blood pressure and thus have cardio benefits? This is not clear in this study – although urea did not cause diuresis (Figure 13), what about other diuretics? And since urea is known to cause diuresis, why did it not work here? – It is important to know if TMAO is very specific in cardioprotection or if many other agents would work as well.

We think that the beneficial (cardioprotective) effect of TMAO is mostly associated with its diuretic effect. We assume that other positive effects of TMAO, such as hypotension and decreased RAAS activity in the heart, were secondary to the reduction of fluid retention in HF animals. TMAO is well-established osmolyte, therefore it is possible that TMAO treatment could also “correct” water distribution between body water compartments.

To fully answer the question of whether the effect of TMAO may be compared to other compounds producing osmotic diuresis (mannitol or gliflozins which increase osmotic diuresis by inhibiting glucose renal reabsorption) is beyond the scope of this study. Additional studies specifically designed to compare and contrast different diuretic agents, including TMAO, would be needed to answer this question.

Here, we compared TMAO to urea because of similarities between the two molecules (both are nitrogenous osmolytes and have a similar molecular weight). As pointed out by the reviewer, in our study TMAO produced significant diuresis whereas urea did not. This was likely due the fact that we evaluated the same (equimolar) doses of urea and TMAO, whereas physiological concentrations of TMAO in plasma is radically lower (micromoles/L) in comparison to urea (millimoles/L). Because of significant differences in plasma concentration of urea and TMAO, considerably higher doses of urea are needed to produce diuresis. Therefore, we think that our study does not contradict the diuretic effect of urea reported by others. Nevertheless, our findings suggest that TMAO exerts a significantly more potent diuretic effect than urea.

We have added discussion on this topic.

c) Results subsection “SD rats”: TMAO rats had higher vasopressin, an anti-diuretic agent. Yet they also had higher diuresis. This seems to be apparent paradox – it should be interpreted in the Discussion, but I cannot find it.

Thank you very much for this remark, we have added discussion on this topic:

“Increased diuresis in TMAO-treated rats was present despite elevated plasma levels of vasopressin which allows the reabsorption of water from the filtrate in collecting ducts. […] This could decrease the reabsorption of water from the filtrate despite the vasopressin-induced increase in the permeability to water of collecting ducts.”

2) Statistical analyses and date presentation:– According to the authors only the (unpaired) t-test was used for statistical comparisons. Data have not been assessed for distribution normality (which would be required) and error bars in Figures 8, 9, 13 suggest non-normal distribution. All data need to be analyzed for normality of distribution, and accordingly parametric or non-parametric tests must be applied.

We have analyzed the data distribution with a Kolmogorov-Smirnov test. This information has been added in Materials and methods subsection, “Data analysis and statistics”. All data were normally distributed, therefore the differences between the groups were evaluated by t-test. All raw data are available as supplementary material attached to the submission.

– Please indicate n number for each group in each figure panel or the corresponding figure legends.

N numbers for each group have been added to figure legends and table legends.

– All P values in Tables 1, 2, and 3 need to be revised and amended, for example, a p<0.23 is meaningless.

This has been corrected.

– Figure 11: X-axis says "Time" but give only single numbers. It is assumed that this is to represent weeks. Please modify.

The figure has been corrected.

3) Language:There are concerns regarding language. Some statements make no sense and need rewording, for example "TMAO reduces mortality in SHHF rats that is associated with diuretic, natriuretic and hypotensive effects. HPS and TMAO are neutral for cardiac LDH.", "Histopathological picture of SHHF heart corresponds", "in SHHF, lungs showed signs of passive hyperemia, with thickening of the interalveolar septa and a weak focal parenchymal edema.", "were euthanized or died spontaneously at the age of 52, 56 and 57-weeks due to hemiparalysis (ischemic stroke), dyspnoe (post-mortem lung edema) and spontaneous death (post-mortem lung edema), respectively." "MAO treated group showed… " "TMAO-treated rats showed a significantly lower diastolic blood pressure and a trend towards lower systolic blood pressure (p=0.01)”, "The general myocardium pattern"… etc. The authors are encouraged to consult with a scientist colleague whose first language is English and, where necessary, improve content, grammar, and syntax.

It has been corrected.

[Editors' note: further revisions were suggested prior to acceptance, as described below.]

Revisions:The main concern is whether the manuscript has been reviewed by a native English speaker, as recommended. Although the language has been improved to some degree, there are still issues ("Water treatment", "Water group") which need correction. For example, the manuscript should always and only refer to a control group (currently "Water") or control (treatment).

According to your suggestions, the manuscript was reviewed by a native English speaker. Additionally, as per your suggestion, we changed the group names to more clearly define each group. Throughout the text, all groups are now defined by their strains, and, within each strain, animals drinking water are defined as “-control,” while animals drinking TMAO solution are defined as “TMAO.” Because throughout the manuscript, we describe three different strains, it is of utmost importance that each strain (SD, SHHF and SD-ISO) and group (treatment vs control) are clearly defined. We thank the reviewer for this comment.

The authors are requested to recheck all data normality using the Shapiro-Wilk test and revise the statistical analyses as required.

We have rechecked all data normality using Shapiro-Wilk test. Differences between the TMAO and control groups were evaluated by Mann-Whitney U test for data that were not normally distributed.

Was the water intake (the TMAO was added into the drinking water) quantified? How was this monitored and how could the authors ascertain the actual plasma TMAO concentration? Also, please indicate how TMAO affected water intake, which will be important for heart failure progression as well.

Water intake was quantified in metabolic cages (Tables 1,2,3 “Survival, Energy and water balance”). In general, at the end of the experiment, the TMAO groups showed higher water intake (however, not significantly higher, p=0.17 for SHHF rats, p=0.9 for SD rats, and p=0.3 for SD-ISO). This seemed to result from a diuretic effect of TMAO, which was found in the present study. In fact, during the first 24-48 hours after the introduction of water containing TMAO, the rats tended to have a lower intake of fluids, likely due to decreased palatability of water containing TMAO. After this time, the intake of the fluid increased and remained stable for the entirety of the experiment (12 months).

Plasma and urine concentrations of TMAO were measured using a Waters Acquity Ultra Performance Liquid Chromatograph coupled with a Waters TQ-S triple-quadrupole mass spectrometer. The mass spectrometer was operated in the multiple-reaction monitoring (MRM) – positive electrospray ionization (ESI) mode. We have several years of experience using this method and evaluating TMAO levels in the plasma and urine (PLoS One. 2017 Dec 13;12(12):e0189310. doi: 10.1371/journal.pone.0189310, Nutrition. 2018 Oct;54:33-39. doi: 10.1016/j.nut.2018.03.004., J Gerontol A Biol Sci Med Sci. 2019 Aug 14:glz181. doi: 10.1093/gerona/glz181., Cardiovasc Res. 2019 Dec 1;115(14):1948-1949. doi: 10.1093/cvr/cvz231.)